# In-built thermo-mechanical cooperative feedback mechanism for self-propelled multimodal locomotion and electricity generation

Xiao-Qiao Wang [1], Chuan Fu Tan[1], Kwok Hoe Chan [1], Xin Lu [1,2], Liangliang Zhu[1], Sang-Woo Kim [3] & Ghim Wei Ho [1,4]

Utilization of ubiquitous low-grade waste heat constitutes a possible avenue towards soft matter actuation and energy recovery opportunities. While most soft materials are not all that smart relying on power input of some kind for continuous response, we conceptualize a self-locked thermo-mechano feedback for autonomous motility and energy generation functions. Here, the low-grade heat usually dismissed as 'not useful' is used to fuel a soft thermo-mechano-electrical system to perform perpetual and untethered multimodal loco-motions. The innately resilient locomotion synchronizes self-governed and auto-sustained temperature fluctuations and mechanical mobility without external stimulus change, enabling simultaneous harvesting of thermo-mechanical energy at the pyro/piezoelectric mechanistic intersection. The untethered soft material showcases deterministic motions (translational oscillation, directional rolling, and clockwise/anticlockwise rotation), rapid transitions and dynamic responses without needing power input, on the contrary extracting power from ambient. This work may open opportunities for thermo-mechano-electrical transduction, multigait soft energy robotics and waste heat harvesting technologies.

---

[1] Department of Electrical and Computer Engineering, National University of Singapore, 4 Engineering Drive 3, Singapore 117583, Singapore. [2] Department of Mechanical Engineering, National University of Singapore, 9 Engineering Drive 1, Singapore 117576, Singapore. [3] School of Advanced Materials Science and Engineering, Sungkyunkwan University (SKKU), Suwon 16419, Republic of Korea. [4] Institute of Materials Research and Engineering, A*STAR (Agency for Science, Technology and Research), 3 Research Link, Singapore 117602, Singapore. Correspondence and requests for materials should be addressed to S.-W.K. (email: kimsw1@skku.edu) or to G.W.H. (email: elehgw@nus.edu.sg)

Untapped thermal energy, especially low-grade heat below 100 °C from various sources namely ambient, industries residual, and non-concentrated solar energy[1–3], is abundant and widely accessible. Indisputably, discharging and dumping of these large quantities of low-grade heat into ambient is considered wasted opportunities, if they are not reclaimed. Despite that, there are huge constraints to extract these valuable low-grade heat using the existing technologies pertaining to the requirements for external temperature difference or temporal heat fluctuation. Hence, dexterous utilization and versatile conversion of the waste heat energy, is now recognized for the sustenance of decarbonized and sustainable ecosystem[4]. Recently, novel material systems and mechanisms have emerged, in attempt to extract and reclaim this prevalent waste heat. One pathway is to convert thermal energy into mechanical work, such as thermo-osmotic energy conversion[5,6] and photochemical/photothermal actuators, which transform solar heat into mechanical deformation[7–13]. Another route is to harvest heat for electricity output, including thermoelectric/thermogalvanic[14–16], pyroelectric transducers[17–19], and water-evaporation-induced generator[20–22]. However, autonomous and continuous extraction and utilization of the low-grade thermal energy from a constant ambient environment for synergistic mechanical and electric energy output is still a grand challenge.

Besides, soft-robotic system has gained increasing attention in recent years, due to its elastic deformability, stimulus responsiveness, and weight/size miniaturization[23–26]. To date, most of the responses are predominantly established from polymer hydrogel[27–29], fluid composite[30], and azobenzene chromophore[7,31,32] with restrictive unidirectional and anisotropic motility i.e., swelling-shrinkage, contraction-expansion, and stagnation-bending. These polymeric gel derivatives, ionic/metal fluids, and photochromophores molecules impose challenges when integrated into consumer-based products, with regard to materials degenerative instability such as fatigue, photobleaching, dry state rigidity etc. Also, except for several light activated self-oscillating materials[33–36], most of these systems rely on or tether to external circuits and/or control devices to regulate or on-off the external stimulus for continuous cyclical motion. That is to say, unless the external stimulus is repeatedly switched on-off, they face limitations in performing repetitive macroscopic motion at high actuation speed under a constant/invariant operation mode. Hence, this underscores the significance to address both the material design and system concept in transposing energy specifically invariant waste/residual heat into diverse mechanical motions, particularly one that is perpetual and self-propelled automation.

In this work, we devise a self-governing thermo-mechano-electrical system (TMES) to exploit low-grade ambient heat for a diverse adaptive mechanical actuation, coupled with thermo-mechanical transducing behavior. The untethered TMES can be simply fueled by a constant low-grade heat source, and self-propelled by an intrinsic built-in thermo-mechanical and mechano-thermal feedback loops, performing perpetual and multimodal locomotions, such as high-frequency translational oscillation, clockwise/anticlockwise rotational, and revolving motor-like motions. Moreover, the locomotion proceeds by self-sustained temperature fluctuations within the ferroelectric TMES, hence allowing synchronous and continuous harvesting of thermo-mechanical energy at the mechanistic principles of pyro/piezoelectric effects. Finally, to meet the foregoing demands of soft actuator that imitates the softness, agility, and self-moving living organisms, a prototypical soft robot termed TMES-bot is sought to reproduce bioinspired self-defensive locomotions with additional power-generating functionality, which can possibly operate in an unstructured outdoor environment.

## Results

**Autonomous TMES design.** The structure design and system concept is presented in Fig. 1a. A thermo-mechanical deformation is first actuated on a hot surface based on the bimorph principle, which requires a bilayer structure with opposite coefficient of thermal expansion (CTE). To fulfill the perpetual motility and thermo-mechanical energy harvesting, materials selection and microstructure design are the essential elements, conferring TMES with mechanical robustness and functional ductility. The TMES mainly consists of a three-dimensionally (3D) aligned ferroelectric polyvinylidene fluoride (PVDF) and polydopamine modified reduced graphene oxide-carbon nanotube layer (PDG-CNT) with nacre-like brick-and-mortar microstructures[37]. The extremely high CTE and mechanical flexibility of PVDF makes it suitable as an active layer in a bimorph thermal actuator[38]. Moreover, the pyroelectric and piezoelectric properties of the ferroelectric $\beta$-phase are the essential prerequisite to harvest thermal and mechanical energy using TMES (Supplementary Fig. 1)[18]. Driven by the thermal responsive bending shape and air-solid (hot surface) temperature gradient, TMES on the flat hot solid surface undergoes a non-equilibrium oscillation, which is followed by simultaneous harvesting of self-created/internalized temperature fluctuations and mechanical actuations within TMES. The 3D alignment was confirmed by the existence of lamellar orientation both on the surface and cross-section of the PVDF (Supplementary Fig. 2), and the PVDF is isotropic in terms of modulus of ~ 1656 MPa (Supplementary Fig. 3), but anisotropic in regards to CTE, showing large transverse thermal expansion and small longitudinal thermal expansion (See details in Supplementary Fig. 4). A direction characterized by anticlockwise angular offset $\alpha$ is defined here as the angle between the longitudinal alignment direction of PVDF and the length direction of TMES strip (Supplementary Fig. 5). Based on alignment control, TMES can form a regular and symmetrical open ring strip and chiral shapes, including various left-handed or right-handed chirality strips (Supplementary Fig. 6) on the hot surface. Notably, the 3D alignment introduces directionality to the otherwise uncontrollable thermal deformation, providing a foundation for realization of autonomous, perpetual and multimodal mechanical locomotions. A TMES film up to 100 cm$^2$ was made (Fig. 1b), which allows easily obtaining of TMES strips at different alignment angles depending on the cutting directions. The other passive layer in the bimorph actuator should be a film with low CTE and electrically conductive to serve as one of the electrodes on ferroelectric PVDF in TMES. In this regard, graphene has a negative CTE $(-8 \times 10^{-6}~K^{-1})$[39] and excellent electrical conductivity. The PDG-CNT synthesized in our work possesses nacre-like brick-and-mortar microstructures (Fig. 1c), and $\pi$-$\pi$ bonding and strong adhesion between PDG and CNT allows PDG-CNT water ink being layer-by-layer deposited on hydrophilic treated PVDF surface without delamination[40], hence ensuring mechanical robustness of TMES to bear large and various deformations. Besides, thermo-mechanical performance optimization and microstructure analysis were done (Supplementary Fig. 7 and 8).

**Thermo-mechanical feedback mechanism for autonomous locomotions.** The autonomous locomotion appears as temperature activated and self-sustained oscillation that synchronizes with the cyclic shifting of center of gravity. TMES at $\alpha = 90°$ (1.5 × 6 cm) was put onto a flat hot surface with PDG-CNT layer facing upwards, and the air temperature was constant at room temperature (~ 23 °C). Owing to the harmonized thermal expansion and mechanical stiffness in combination with precise alignment direction control in the microstructured PVDF, TMES

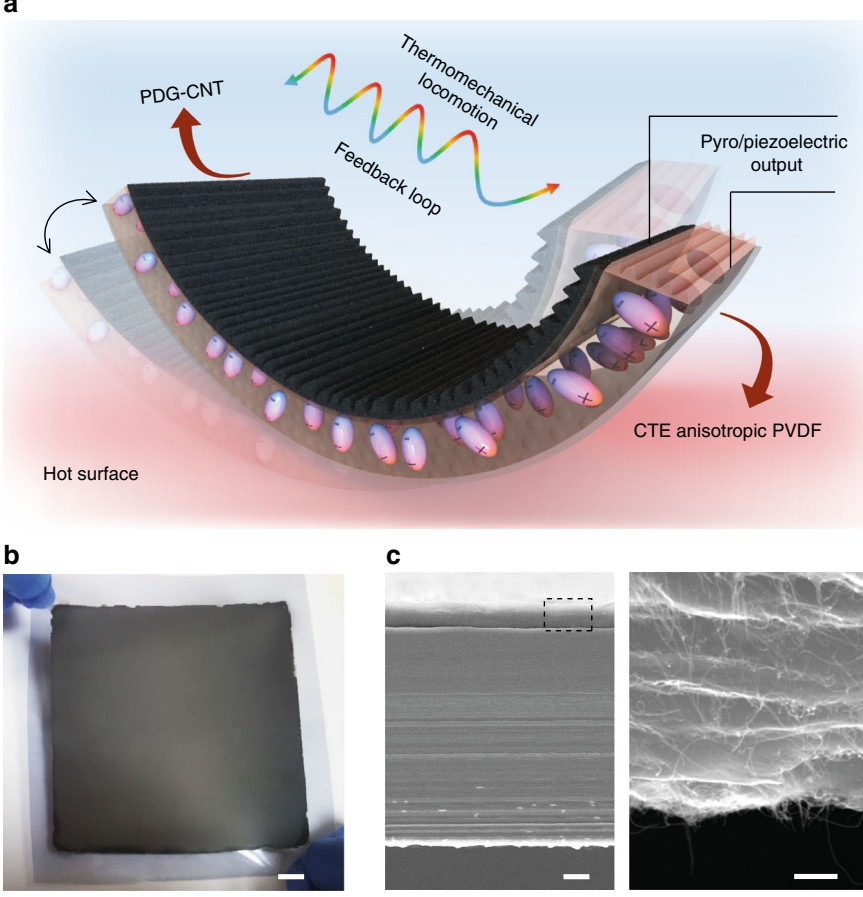

**Fig. 1** The structure design and system concept of TMES. **a** Schematic illustration of the design concept of thermo-mechano-electrical conversion based on a bimorph actuator. **b** Photograph of a large-area TMES sample. **c** Cross-sectional scanning electron microscopy (SEM) image of PVDF/PDG-CNT bimorph (left) and magnification of PDG-CNT nanocomposite layer (right). The scale in **b**, left and right of **c** are 1 cm, 10 μm, and 200 nm, respectively

was successfully actuated with regular bending deformation. The images were taken and analyzed by fitting the maximum bend radius of the film with a circle (Fig. 2a and Supplementary Fig. 9a). The curvature increases nearly linearly from 0.36 to maximum $0.94\ cm^{-1}$ when the surface temperature increases from 30 to 65 °C (Fig. 2b). The infrared images show changes in curvature and temperature distribution along the length direction of the film. Typically, from the middle of the film in contact with hot surface to the sides, the temperature of TMES gradually attenuates, and thus the central contact point exhibits the maximum bending curvature. Notably, TMES can generate visual bending deformation even at a very low temperature of 27.2 °C, revealing high sensitivity of the thermo-mechanical response. When the surface temperature is increased to 50 °C, an intriguing autonomous and discontinuous weak oscillation behavior of TMES transpires. Further increase in temperature even by one degree up to a critical temperature of 55 °C, switches TMES into a fast continuous and dramatic oscillation mode (Supplementary Fig. 9b). Detail investigation shows TMES undergoes a fast bending deformation within 5 s, before triggering the oscillation behavior. Then the oscillation amplitude quickly maximizes in the following 9 s and stays constant thereafter (Supplementary Movie 1). Steady-state motion of TMES at a constant temperature in the range of 55–65 °C, turns out to be a continuously repeated mechanical oscillation in two-dimension (2D), accompanied by shifting of center of gravity without obvious macroscopic bending curvature variations.

To carry out kinematic and mechanical analyses, we employed a fluorescence labeling and tracking method, to determine the oscillation trajectory, amplitude and speed, as well as to track the position of center of gravity. The cross-sections of TMES in length direction were marked with fluorescence points, and the oscillation process was exposed to a 365 UV light and recorded by a high-speed camera at 240 fps. In this way, motion of each fluorescent points on TMES in 2D during the oscillation process can be precisely located with a time resolution of ~ 4.2 ms and displacement resolution of ~ 0.18 mm (Fig. 2c and Supplementary Movie 2). The trajectory of these fluorescent points during a complete oscillation cycle from the leftmost to rightmost and then back to leftmost position, at four different temperatures, 50, 55, 60, and 65 °C were plotted and selectively shown in Fig. 2d (from bottom to top). At 50 °C, when weak oscillation occurs, the middle point on TMES, point 4 is virtually at a standstill and remains contacted to the hot surface, whereas the points on the sides of TMES swing slightly. At 55 °C, when continuous and intense oscillation is maintained, point 4 is seen to separate from the surface during the oscillation motion, and the 2D trajectories depicted are relatively symmetrical. The oscillation asymmetry arises when the surface temperature was further increased. The trajectories of point 4 at both 60 and 65 °C demonstrate larger maximum displacements along both the $x$ and $y$-axis on the right sides compared to that of the left sides, suggesting that the oscillation amplitude of TMES has a specific propensity in terms of direction at high temperatures. This directionality offers a

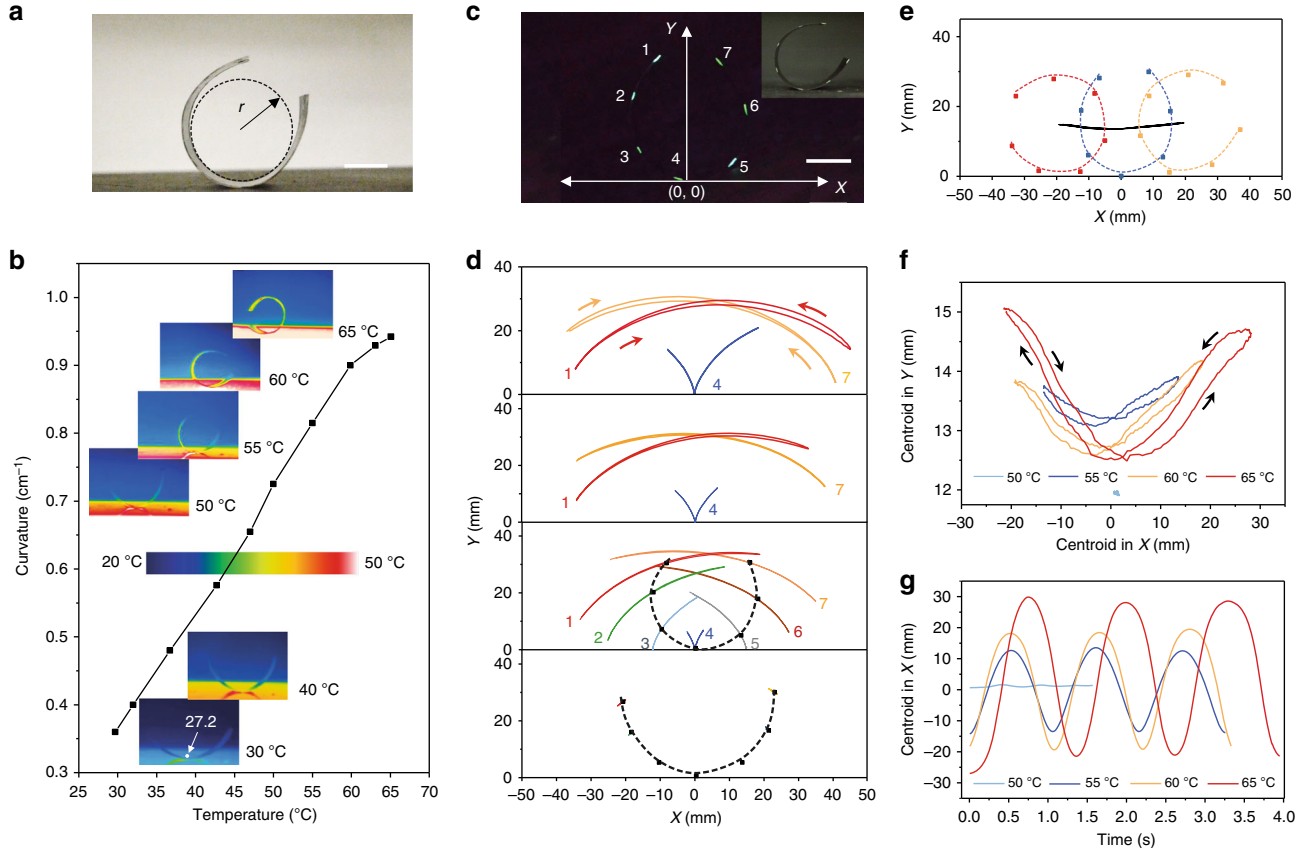

**Fig. 2** Autonomous thermo-mechanical oscillation, kinematic tracking, and mechanical analysis. **a** Photograph of TMES on a hot surface at 60 °C. **b** Infrared images and curvature change of TMES on the hot surface with different temperatures. **c** Fluorescent image of marked TMES in UV light and its 2D coordinate system. The inset is a photograph of marked TMES in visible light. **d** The 2D trajectory of marked points on the cross-section of TMES during a complete oscillation cycle from the leftmost to rightmost and then back to the leftmost position, at four different temperatures, 50, 55, 60, and 65 °C (from bottom to top). **e** Plot of center of gravity during a complete oscillation cycle based on trajectory of the marked points. **f** The 2D trajectory of center of gravity at different temperatures. **g** Time-dependent center of gravity in the *x*-axis during three consecutive oscillation cycles at different temperatures. All scale bars correspond to 1 cm

feasibility not only to devise a more sophisticated but also a directional controllable motions driven by the mechanical oscillation. In terms of oscillation amplitude, the tip displacement from the leftmost to rightmost position (point 1) in the *x*-axis direction reaches 52.1, 67.2, and 79.5 mm, at 55, 60, and 65 °C, respectively, demonstrating increasing oscillation amplitudes at higher temperatures. Besides, we note that the trajectories of the points on sides of TMES are loop curves, especially for the two tip points 1 and 7. The curve of point 1 from the leftmost to rightmost is under the curve from the rightmost to leftmost along the *y*-axis, whereas the opposite case is illustrated in point 7 as marked at the top of Fig. 2d. These motion characteristics inspire us to further explore and explain the intrinsic mechanism that materializes a self-driven oscillation. In addition, a TMES sample in square shape can oscillate in a similar fashion as that of TMES strip in the same length at 60 °C as shown in Supplementary Fig. 10. On the basis of successful kinematic tracking of the fluorescence points, centroid analysis is further employed to locate the position of center of gravity (Fig. 2e and Supplementary Fig. 11), thus enabling unequivocal explanation of the oscillation in a mechanical perspective. The trajectories of center of gravity of TMES corresponding to the oscillation traces at different temperatures are shown in Fig. 2f. The movement direction and amplitude of center of gravity determine the motion characteristics. For example, a displacement of center of gravity from the left to the right will result in a torque that drives TMES to move

in the same direction on the flat hot surface[10]. Figure 2g shows the shifting of center of gravity along the *x*-axis during three consecutive oscillation cycles at different temperatures. The oscillation frequencies are ~55, 54, and 46 min$^{-1}$, respectively, at 55, 60, and 65 °C. Remarkably, we monitored the motility of TMES at 55 °C over 10 h as shown in Supplementary Fig. 12 and Supplementary Movie 3, and repeated oscillations over 33,000 times which displays stable and ceaseless/unrelenting motion. Unlike any conventional actuation subjected to cyclic mechanical deformations, the oscillation of our TMES is characterized by shifting of center of gravity at relatively low temperatures, which circumvents repeated deformations and materials deterioration, thereby conceptualizing a non-fatigue perpetual motility.

To this stage, we are able to propose a crucial thermo-mechanical interaction principle that activates and sustains the oscillation. Owing to the uniform bimorph structure without any gradient thickness design in both PDG-CNT and PVDF, the thermoresponsive bending deformation of TMES appears to be symmetrical with an elliptical cross-section, where the maximum curvature occurs in the middle of length direction. Thus TMES is able to keep its equilibrium on the hot surface at temperatures below 55 °C. In principle, without any other stimulus from external fields, TMES should maintain its balance, even when much larger deformations are produced at 55 °C and above. Seemingly, TMES works on the basis of interfacial temperature between the hot surface and the cool ambient air (23 °C), thus

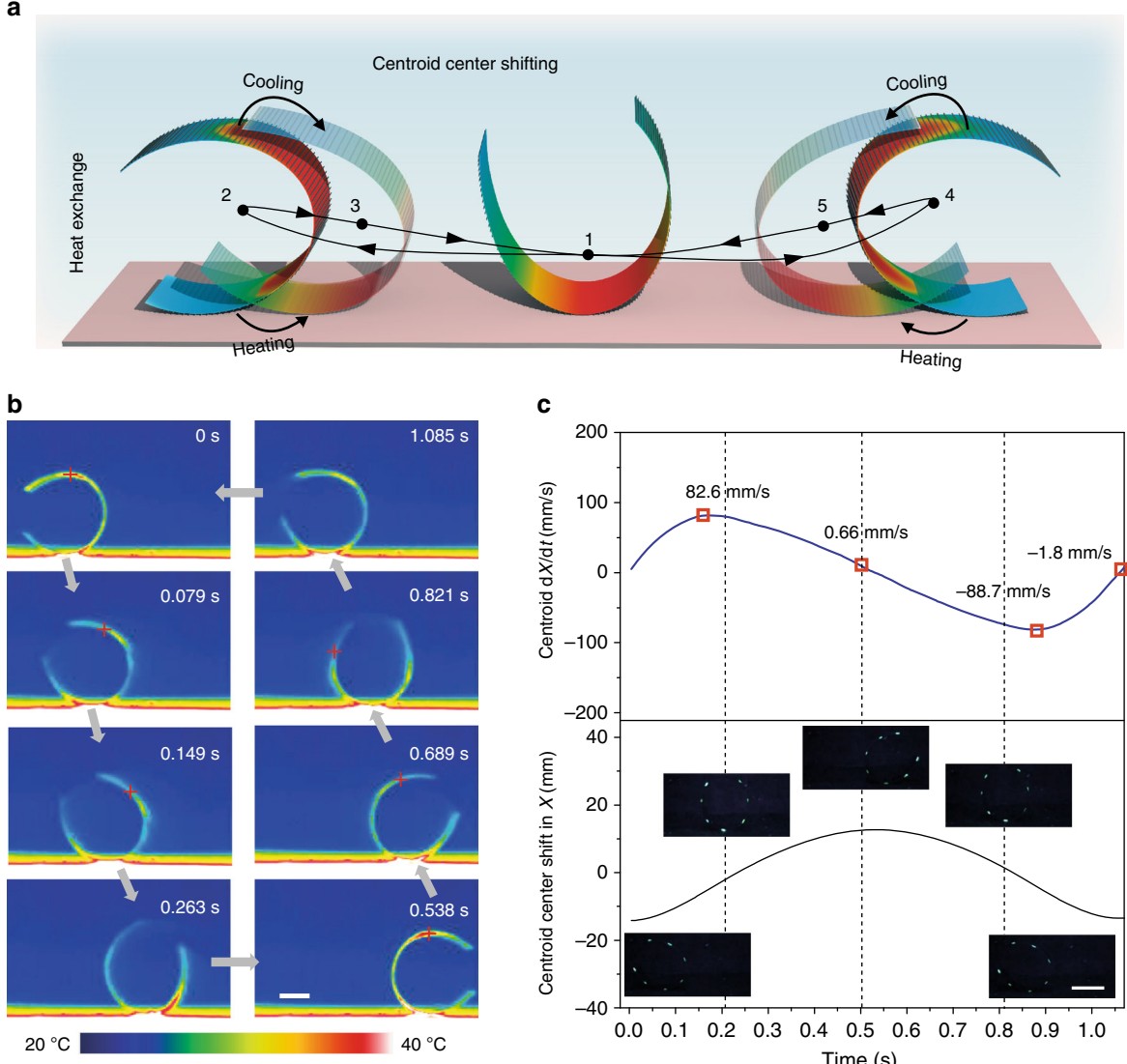

**Fig. 3** Thermo-mechanical feedback mechanism for the autonomous oscillation. **a** Schematic illustration of the heat exchange followed by center of gravity shifting of TMES during the oscillation process. **b** Time-dependent infrared images. **c** Centroid and rate of change of centroid of TMES in the *x*-axis during a complete oscillation cycle at 60 °C. Insets of **c** are fluorescent images of marked TMES corresponding to the oscillation positions. All scale bars correspond to 1 cm

exposing to a temperature gradient that attenuates from hot surface to above air (Supplementary Fig. 13). The color distribution in TMES illustrated in Fig. 3a corresponds to the thermal distribution (Supplementary Fig. 14 and Supplementary Movie 4). At 55 °C, TMES is in a highly unstable state when subjected to the large deformation, which tends to be contradictorily unfold owing to the cold air, especially at the left and right tips of TMES in length direction which have the lowest temperature. In this case, a little curvature difference between the two tips caused by unfolding desynchrony will produce geometric asymmetry, which consequently breaks its initial balance and activates the oscillation. Once initiated, TMES first falls onto the left as illustrated in Fig. 3a, along with a continuous shifting of the center of gravity, which brings TMES to the leftmost position. Next, the left side of TMES is now in contact with the hot surface and heated up, while the right side exposed to air is cooled down. This combined effect leads to distinct geometrical shape with center of gravity shifting back to the right. Thermal traces in Fig. 3b show that temperature on the right side continuously decreases during 0 to 0.149 s in this process. Due to the motion

inertia, TMES will cross the middle position and continue to oscillate to the rightmost position, and similar heating and cooling effects to that of the leftmost position will happen, thus bringing TMES back to the leftmost position. This as well explains the trajectory loops of the tip points as discussed above, since the curvature of the tips keeps decreasing once it oscillates away from the hot surface until the next oscillation cycle starts. Temperature profile of marked points on TMES is additionally analyzed (See details in Supplementary Fig. 15). Center of gravity shift analysis in Fig. 3c shows that TMES attains the highest moving speed of over 80 mm/s along the *x*-axis at around the initial middle position, while the lowest speed appears at the leftmost and rightmost position. Therefore, the fast heat exchange at the two positions is essential to reversing the moving direction of the center of gravity, thereby maintaining fast continuous oscillation cycles. Taken together, a thermo-mechanical and mechano-thermal feedback loop is indigenously established by the self-regulated bending deformation and continuous auto-displacement of center of gravity of TMES, which in turn sustains the oscillation through the alternative heating/cooling process.

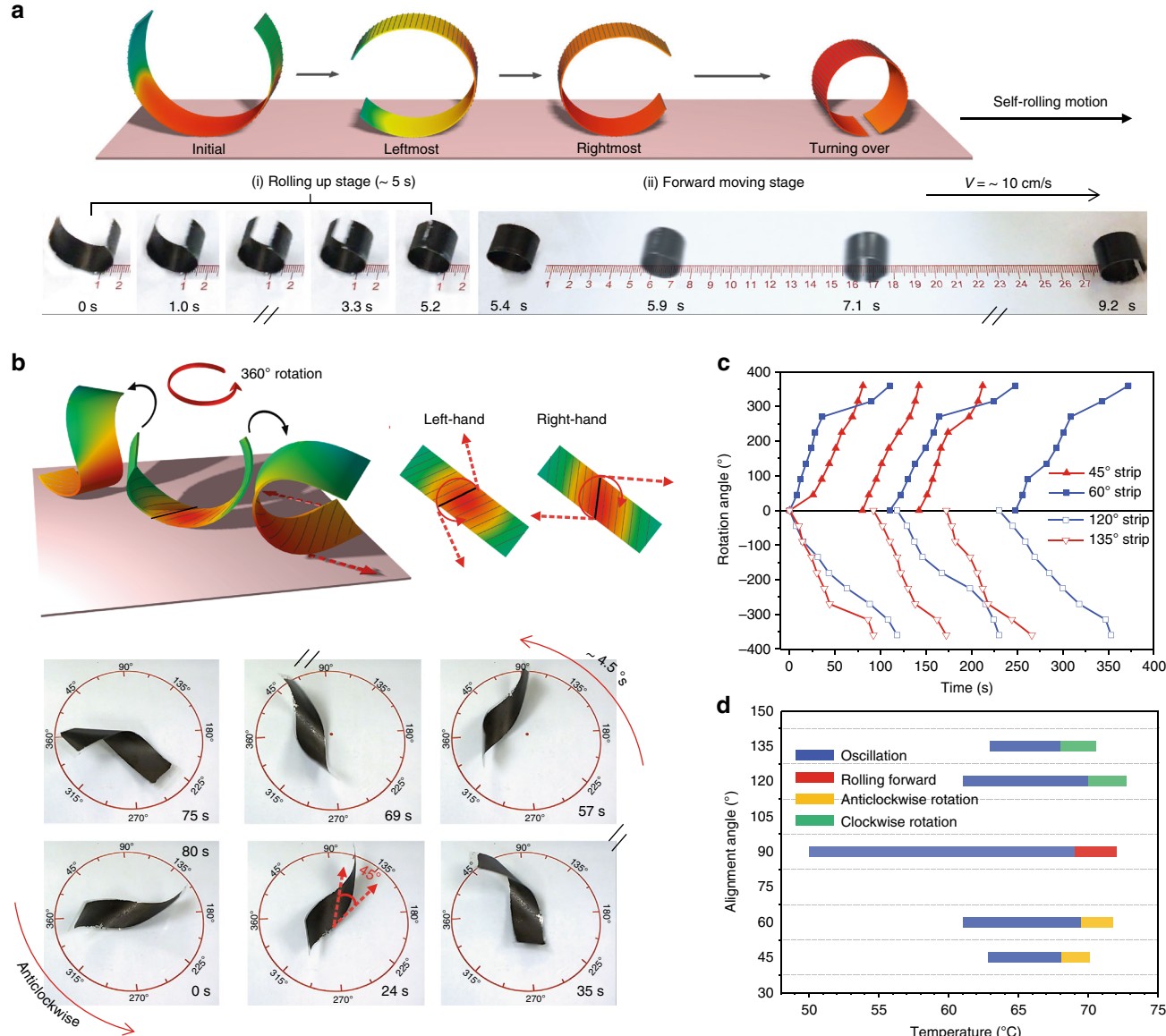

**Fig. 4** Realization of self-propelled multimodal locomotions. **a** Schematic representation and snapshots of a self-rolling motion of TMES at $\alpha = 90°$ on a hot surface at 70 °C. **b** Schematic illustration of self-rotation movement with direction determined by the alignment angle of PVDF in TMES, and snapshots of TMES at $\alpha = 45°$ during an anticlockwise rotation process. **c** Time-dependent rotation angle of TMES at four different alignment angles during three consecutive 360-degree rotation cycles. **d** The locomotion modes of TMES strips at different alignment angles on the hot surface with temperatures ranging from 50 to 75 °C

**Self-propelled multimodal locomotions**. We further explore other motion modes that can be triggered by the basic oscillation motion. After several trials, we found that anomalous directional self-rolling motion can be realized using a single strip of TMES at $\alpha = 90°$. Similar to the previous, TMES assumes an elliptical cross-section with curvature differences along the length direction at temperature < 70 °C and displays oscillatory response. However, when the hot surface temperature is ~ 70 °C, oscillations with increased speeds and amplitudes enable cyclic and alternative contact of the two sides of TMES with hot surface, which is followed by a decreased temperature gradient range in TMES as schematically illustrated in Fig. 4a. With increasing curvatures on the left and right sides, the gap between the two ends of TMES keeps reducing and finally disappears. As such, TMES rolls up into a perfect cylinder having a complete circular cross-section. Consequently, TMES turns over and works as a fast rolling-forward alike a revolving motor that is self-propelled by the last

oscillating shift of center of gravity. The part 1 of Supplementary Movie 5 shows the rolling motion. The autogenous rolling up process completes within 5.4 s before rolling motion starts with a speed of ~ 10 cm/s (Fig. 4a). Moreover, the rolling direction is not arbitrary but controllable, owing to the propensity of oscillation amplitude as previously discussed. In addition, inverted-series-connected bimorph actuator[41] consisting of two identical TMES strips was assembled, and exhibits similar oscillation driven rolling motion, favorably with a wider working temperature range from 65 to 80 °C (See details in Supplementary Fig. 16, 17 and part 2 of Supplementary Movie 5), demonstrating controllability of the rolling dynamics.

To uncover other mechanical actuation capabilities, chiral shapes TMES are further studied. It is found that TMES at $\alpha = 45°$, 60°, 120°, and 135° exhibit similar oscillation behaviors to that of TMES at $\alpha = 90°$ at hot surface temperature above 60 °C, except that different inclined oscillation directions corresponding

to the twisting angles are observed. Interestingly, clockwise or anticlockwise self-rotation movement are triggered depending on the twisting chirality when the hot surface temperature reaches ~70 °C. As illustrated in Fig. 4b, the actuated TMES in a twisted configuration oscillates in the direction perpendicular to the middle twisting axis. As such, every strong oscillation produces a torsional force acting on the middle axis, thereby exerting a rotational kinetics. When TMES ($\alpha = 45°$ and 60°) twists in a left-handed mode, the oscillation induced torsion yields anticlockwise rotation; while the right-handed oscillations (TMES at $\alpha = 120°$ and 135°) produce clockwise rotations. Supplementary Movie 6 and 7 show the anticlockwise and clockwise rotation process of TMES at $\alpha = 45°$ and $\alpha = 120°$, respectively, and each of the 360-degree rotation is accomplished within 80 s (~ 4.5°/s) and 120 s (~3.0°/s) (Fig. 4b and Supplementary Fig. 18). The 360-degree rotational motion retains its stability over consecutive cycles for each of the twisted TMES (Fig. 4c).

Briefly, two parameters that determine the autonomous locomotion mode of TMES strip ($1.5 \times 6$ cm)—the alignment angle and the hot surface temperature (Fig. 4d). TMES at $\alpha = 90°$ under bending deformation achieves atypical translational locomotion, including self-oscillation in a wide temperature range from 50 to 70 °C and self-rolling motion at 70–72 °C. TMES at $\alpha = 45°$ subjected to left-handed twisting deformation generates rotational locomotion, including self-oscillation from 63 to 68 °C and anticlockwise self-rotation from 68 to 70 °C; TMES at $\alpha = 120°$ in right-handed twisting mode produces self-oscillation from 61 to 70 °C and clockwise self-rotation from 70 to 72 °C. TMES twisted in enantiomer shapes ($\alpha = 45°$ versus $\alpha = 135°$, and $\alpha = 60°$ versus $\alpha = 120°$) work in a similar motion mode but opposite movement direction in the temperature range. In addition, changing the length of TMES will affect the working temperature ranges of different modal locomotions. For example, if the length of TMES strip at $\alpha = 90°$ decreases from 6 to 4 cm, activation temperatures of its self-oscillation and rolling motion should be higher than 50 and 70 °C, respectively. The thermo-mechanical feedback triggered motion proposed here represents a general concept for bi-layered actuators or thermo-mechanical responsive materials, and it is possible to design various autonomous locomotions working at different temperature ranges based on different thermo-mechanical response sensitivities of the materials systems.

Apart from the multiple locomotion modes, TMES can also be utilized as a thermo-mechanical engine that performs several exclusive mechanical functions. Supplementary Movie 8 part 1 shows that, a single strip of TMES ($\alpha = 90°$) tightly grasps a cargo and performs equitable mechanical oscillations on the hot surface at 65 °C compared to that of unloaded film. Autonomous cargo transportation across a hot surface at 70 °C at a speed of ~ 6.7 cm/s is also demonstrated in Supplementary Movie 8 part 2. In addition, a thermo-mechanical see-saw balance is shown in Supplementary Movie 8 part 3.

**Thermo-mechanical locomotion energy harvesting**. More significantly, the thermo-mechanical energy can be synchronously harvested using the monolithic TMES. The PDG-CNT layer serves as the top electrode on PVDF ($\alpha = 90°$), the other side is coated with thin poly(3,4-ethylenedioxythiophene)-poly(styrenesulfonate) (PEDOT:PSS, ~ 1 μm) as the bottom electrode, and carbon fibers are finally attached to the two faces at the edge. Upon placing on the hot surface, the TMES generator undergoes continuous mechanical oscillations that synchronize with heat fluctuations within TMES, thereby resulting in a cooperative thermo-mechano-pyro/piezoelectric effect (Supplementary Fig. 19a). Kinematic trajectory of the leftmost point on TMES

generator ($1.5 \times 6$ cm) was recorded to track its motion, and the output open circuit voltage ($V_{oc}$) and short circuit current ($I_{sc}$) were collected (Supplementary Fig. 19b). Figure 5a shows continuous and stable $V_{oc}$ and $I_{sc}$ output during five consecutive oscillation cycles at 60 °C, and peak $V_{oc}$ and $I_{sc}$ generated are ~ 24 V, and ~ 47 nA, respectively. It is observed that two $V_{oc}$ peaks appear during one oscillation cycle, which are attributed to two positional switching and distinct thermal heating/cooling effect acting on TMES (Supplementary Movie 9). As shown in Fig. 5b, the $V_{oc}$ continuously increases, when TMES stays almost stationary in the leftmost position while continuous air cooling effect acts on the right side of TMES. The cooling results in a continuous increase of $V_{oc}$ until fast switching of TMES from the leftmost to rightmost position reverses the cooling to heating effect. Correspondingly, $V_{oc}$ gradually decreases and the first $V_{oc}$ peak emerges. After a short heating effect due to heat transfer from the hot surface to the right side of TMES, air cooling effect takes on the left side of TMES, thereupon $V_{oc}$ keeps increasing accordingly until TMES switches from the rightmost to leftmost position, and the second $V_{oc}$ peak appears. Therefore, the heating/cooling pyroelectric effect mainly contributes to $V_{oc}$ output while locomotions sustain the essential cyclic thermal fluctuations. In contrast, $I_{sc}$ peaks correspond to each of the oscillation motions (Supplementary Movie 10), suggesting that mechanical piezoelectric effect dominates the peak $I_{sc}$ output. We further recorded the kinematic trajectories and electric signals at 55 and 65 °C (Supplementary Fig. 20), and compared the average cycling time, maximum displacement, position switching speed (Supplementary Fig. 21) and peak $I_{sc}$ at different temperatures. It is revealed that the cycling time largely increases with increasing temperature, and thus the frequency of electric signal output decreases (Fig. 5c). Whereas the peak $I_{sc}$ increases owing to larger displacement and higher motion speed at high temperatures. A TMES generator with size of $6 \times 6$ cm² at 60 °C can generate peak $V_{oc}$ of ~ 67 V and $I_{sc}$ of ~ 145 nA (Supplementary Fig. 22). To better understand the TMES, the energy conversion process and efficiency are discussed (See details in Supplementary Note 1). In the TMES developed here, the temperature difference between waste heat and air triggers mechanical deformation of TMES, and also sustains the continuous locomotions and temperature fluctuations. The mechanical locomotion stress and temperature gradient are correspondingly harvested as piezoelectricity and pyroelectricity based on the ferroelectric property of PVDF. Therefore, our TMES opens up a new perspective to automatically extract pyro/piezoelectric energy from a constant environment ingeniously utilizing self-sustained mechanical motions and temperature fluctuations in built within the soft energy generators.

Next, we fabricated a soft system named TMES-bot to imitate living organism behavior (Fig. 6a). Throughout the animal kingdom, anti-predator adaptations are mechanisms developed through evolution to assist the prey to fend against predator. The adaption pathways are often deterministic and can rapidly transit from one type to another, depending on the predicaments. The first line of defense is to avoid detection, through shrinking or curling up mechanism. Next, the prey may ward off attack by making known their presence of strong defence using writhing and oscillating motion to startle or signal to its predator that a pursuit is not worthwhile. Finally, when the predator is distracted, the prey makes an unnoticeable escape by rolling off, via altering its shape to generate a propulsive force[42]. It is known that several species of elongated organisms can transform their bodies into a loop or spherical posture to roll, such as certain caterpillars, Armadillidiidae, mantis shrimp, pangolins, desert wheel spiders, and hedgehogs[42,43]. In our case, we use the TMES-bot to draw similar biological analogies of sequential 'self-defense' actions.

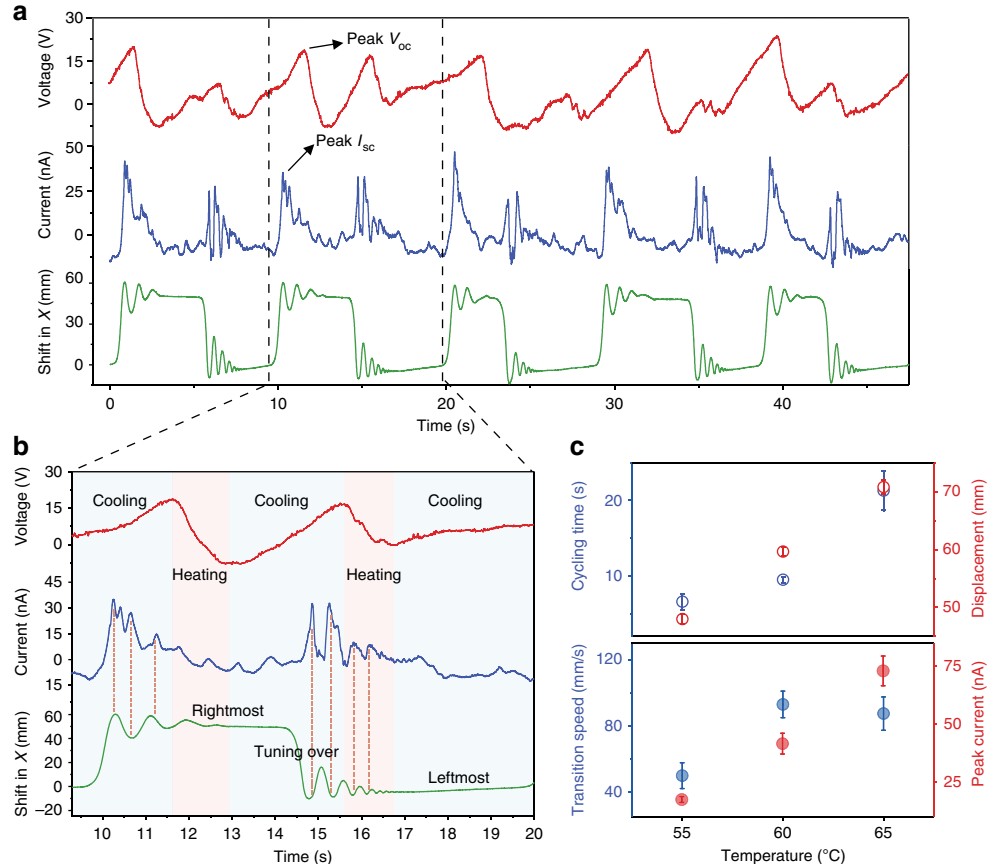

**Fig. 5** Thermomechanical locomotion energy harvesting utilizing the pyro/piezoelectric effect. **a** Time-dependent shifting in the *x*-axis, short circuit current and open circuit voltage of TMES ($\alpha = 90°$) during five consecutive oscillation cycles on the hot surface at 60 °C and **b** magnification in one oscillation cycle. **c** Average cycling time, maximum displacement, transition speed, and peak $I_{sc}$ at 55, 60, and 65 °C. Error bars represent SD

When TMES-bot inadvertently topples over and lands on its back on a hot ground, it has to muster its self-defense skill or otherwise succumbs to predation. At first, the TMES-bot will instinctively fold its body akin to armouring itself with protective hard outer shell/exoskeleton. Within seconds, its body can be curled up, achieving the center of gravity instability, which subsequently triggers fast oscillating motions, analogous to writhing on its back. With increasing oscillation amplitude, TMES-bot will further reform its body into a wheel-like shape/posture, and finally extricate itself by successful flipping and rolling motions (Fig. 6b and Supplementary Movie 11). Working as an active locomotive-adaptive system, the TMES-bot alters autonomously for the purposes of assimilating built-in defense to reproduce the aforementioned anti-predator adaptation pathways. More interestingly, our TMES-bot integrates a thermo-mechanical energy harvesting component, and energy extraction process operates during the oscillation locomotion utilizing pyro/piezoelectric effect (Fig. 6c). The TMES-bot successfully charged an external storage capacitor to 5 V, and lighted a red LED (Fig. 6d). Finally, we demonstrate feasibility of TMES to perform sunlight photothermal to mechanical work and electricity conversion under an ambient outdoor environment. The outdoor time-dependent solar intensity was monitored while the corresponding ambient temperature and surface temperature of a black tiled pavement that was naturally heated up by sun were recorded (Supplementary Fig. 23). It is apparent that TMES shows effective oscillations under non-concentrated and constant sunlight irradiation (Supplementary Movie 12). The peak $V_{oc}$ and $I_{sc}$ output of TMES ($6 \times 6$ cm²) show ~ 45 V and ~ 135 nA basically driven by invariable sunlight stimulus (~ 85 mW/cm²) (top of

Fig. 6e). Besides, under sunlight at a relatively low intensity (~ 50 mW/cm²), a selective focused sunlight directed by a lens can also successfully trigger mechanical oscillations after breaking the initial symmetric thermal distribution in TMES (Supplementary Fig. 24), illustrating the concept of solar enabled photoswitch to mechanical works (bottom of Fig. 6e). Such multi-functional TMES may provide new opportunities for developing biomimetic robotic functionality that is free from the traditional need of powering systems, and conversely offer an on-site waste heat energy recovery strategy.

## Discussion

Our results demonstrate the use of TMES that opens up a promising avenue of smart soft material system that operates on self-regulated thermo-mechano feedback—based on structural instability that readily transit between various steady states. The monolithic TMES is capable of processing autonomous extraction of thermal energy from a constant low-grade heat environment and converting it to diverse locomotions and energy generation functions. Kinematic tracking, mechanical analysis, and dynamic thermal imaging disclose the essential principles that activate and maintain the ceaseless thermo-mechanical synchronized mechano-thermal feedback loop, consequently generating adaptive locomotions and cooperative pyro/piezoelectricity. Such soft material system has shown to effectively interface with low-grade ambient heat and adapt to unstructured environment, while not limited by the dependence on electrical or pneumatic tethers. Ultimately, the self-locked thermo-mechanical feedback mechanism described here may offer new possibilities for

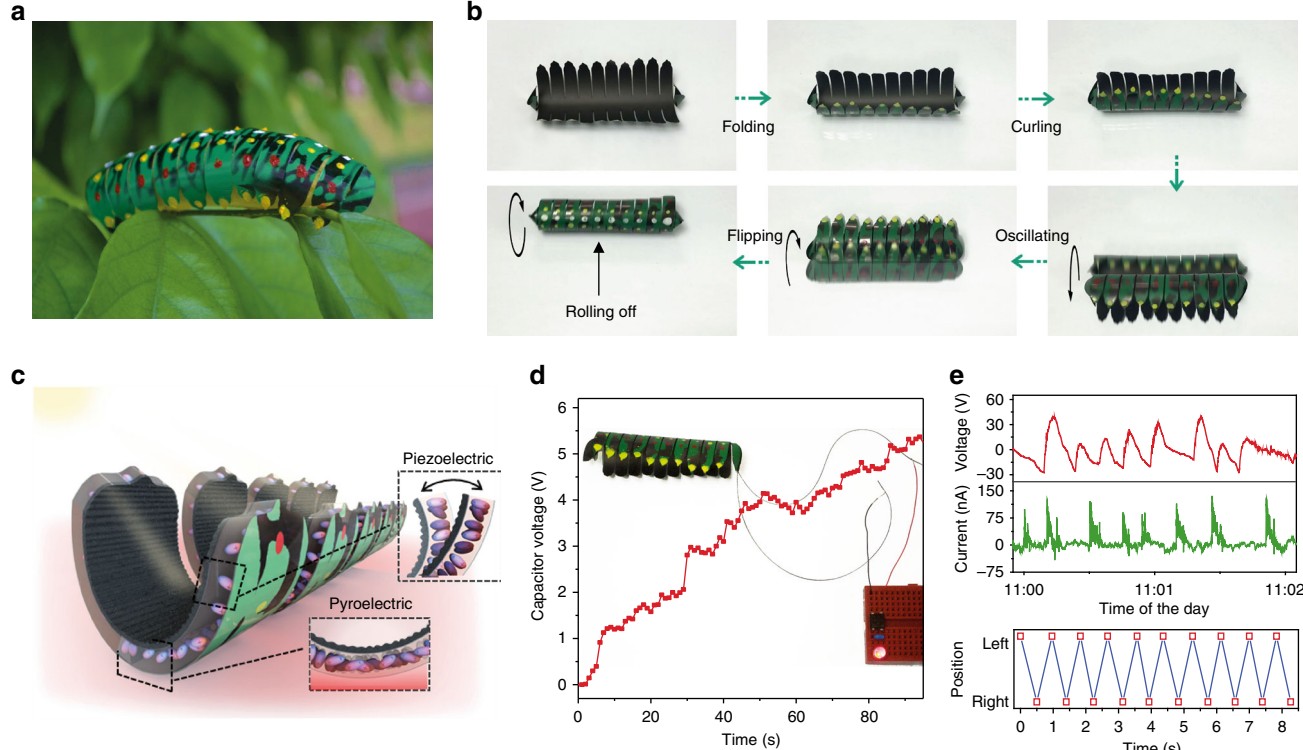

**Fig. 6** Power-generating soft TMES-bot imitating self-defensive living organism behaviors. **a** Photograph of the fabricated TMES-bot. **b** Active locomotive evolution of TMES-bot for self-defense purpose. **c** Schematic showing pyro/piezoelectric effects of the power-generating TMES-bot. **d** Charging characteristic of TMES-bot and an inset photograph of a lighted LED using a charged capacitor. **e** Open circuit voltage/short circuit current generated by locomotion of TMES on a tiled pavement under an ambient outdoor environment (top) and photothermal-directed cyclic oscillations of the TMES (bottom)

autonomous multigait soft energy robotics and waste heat harvesting technologies with versatile thermo-mechano-electrical transduction.

## Methods
**Fabrication of the TMES**. Graphene oxide (GO) was prepared according to the modified Hummers method. Multiwalled carbon nanotube was purchased from Nanjing XFNANO Materials Tech Co., Ltd. Dopamine hydrochloride and poly (3,4-ethylenedioxythiophene)-poly(styrenesulfonate) (PEDOT:PSS) was purchased from Sigma-Aldrich. Poled 3D aligned polyvinylidene difluoride (PVDF) with a thickness of 80 μm was purchased from Fils Co., Ltd company. Typically, 0.1 g of GO was dispersed in 20 ml of deionized water and sonicated into GO dispersion. Then the GO dispersion and 0.05 g of dopamine hydrochloride were added into the 200 ml of Tris buffer solution (pH 8.5), followed by sonication in ice bath for 15 min. The mixture solution was stirred vigorously at 60 °C and maintained for 24 h. The obtained PDG solution was centrifuged at 14,000 rpm and washed with deionized water for three times. After that, CNT at weight ratio of 1:9 to PDG was added to the PDG solution and stirred for 2 h. The PDG-CNT mixture solution was then put in a programed 2 s on and 5 s off ultrasonication treatment in ice bath for 30 min using a 20 kHz sonicator at 200 W, forming the PDG-CNT water ink (2 mg/ml). Both sides of the 3D aligned PVDF film was spin-coated with thin layers of hydrophilic PEDOT:PSS (~ 1 μm), and annealed on a hot plate at 65 °C for 1 h. Finally, the PDG-CNT ink was cast on the PVDF and dried at 40 °C, and this process was repeated for several times until the desired thickness was achieved. The TMES-bot is fabricated by cutting and patterning a TMES film using Silhouette CAMEO, and is then painted using pigmented acrylic ink.

**Characterization and measurement**. Morphology and configuration of TMES were studied by field-emission scanning electron microscopy (FESEM, JEOL FEG JSM 7001F). Crystallographic information on the 3D aligned PVDF were obtained using X-ray diffraction (XRD, Philips X-ray diffractometer with Cu Kα radiation at λ = 1.541 Å). Thermal expansion of 3D aligned PVDF along the transverse and longitudinal direction were studied using an Olympus U-SPT optical microscope in reflection mode. Temperature of the hot surface was controlled by a Eurotherm heater (230 V, 50 Hz). For motion tracking, oscillation process of the fluorescence labeled TMES was exposed to a UV light at wavelength of 365 nm, and captured by an iphone 8 camera in slow motion mode (240 fps). The obtained video was

processed and analyzed by ImageJ. The infrared images and videos were captured and analyzed using a FLIR E50 infrared camera (30 fps) and FLIR Tools+, respectively. The output of short circuit currents and open circuit voltages during the thermo-mechanical locomotion process were measured by connecting the two electrodes of TMES to a low-noise current preamplifier (MODEL SR570) and an electrometer (KEITHLEY 6517B), respectively, and the results were recorded by an oscilloscope (Tektronix DPO 7254). A commercial 100 nF capacitor integrated with a rectifying bridge circuit was used to store the thermo-mechanical locomotion energy and then light up a red LED. The outdoor measurement was done on the rooftop of a three-storey building in NUS, Singapore, on 24 January 2018.

**Data availability**. The data that support the findings of this study are available from the corresponding authors upon reasonable request.

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

## Acknowledgements

This work is supported by the National Research Foundation Singapore, Ministry of National Development (MND), R-263-000-C22-277, and NUS Hybrid-Integrated Flexible (Stretchable) Electronic Systems Program grant number R-263-501-011-731.

## Author contributions

G.W.H. proposed the research direction and supervised the project. X.Q.W. and G.W.H. conceived and designed the experiments. X.Q.W., K.H.C., X.L. and L.Z. fabricated the devices and performed the experiments. X.Q.W., C.F.T., G.W.H., and S.-W.K. collected, analyzed the data and developed the explanation. G.W.H., X.Q.W., and S.-W.K. wrote the manuscript. All authors discussed the results and reviewed the manuscript.

## Additional information

**Competing interests:** The authors declare no competing interests.

