## [Peer Review File · Nature Communications]

Reviewers' comments:

Reviewer #1 (Remarks to the Author):

This manuscript reports an interesting development of thermo-mechano-electrical system (TMES) using a multilayered soft material structure. This device is able to perform perpetual and multimodal locomotions including translational oscillation, rotational and revolving when placed on a hot surface. This heat-induced motion is further demonstrated for electric energy generation and generating bio-mimic responses. This work is innovative and demonstrates a new way of using low-level thermal energy to achieve multiple intriguing functions. The heat-induced motion mechanism is also analyzed in great details. This work possesses a high level of novelty and quality for a publication on Nature Communications. However, there still are several issues that need to be addressed.

1. The motion of TMES was attributed to the temperature and heat transfer. The results at different surface temperature was shown and compared. Heat transfer is also related to the temperature gradient or the temperature difference. This may be largely related to how fast the ring oscillates or how fast it moves. Therefore, temperature difference between different points on the device also need to be analyzed and correlate to the motion patterns.
2. Following above question, the smaller the ring is, less temperature gradient there is. Can the authors comment and predict how this behavior changes as the size of the ring reduces? What would be the balance between the temperature and the ring size?
3. The fluorescence trace image shown in fig. 2c is confusing. Considering the ring was rolling back and forth, the lateral positions of the fluorescence spots should constantly change and this change should be more obvious for upper spots. Why in the image, all the fluorescence spot traces seem following the same profile? The device profile also need to be provided in the image to reveal the relative position changes.
4. In the energy harvesting part, the authors demonstrated a good match between the x position and current output. However, the voltage profile appears rather irregular. For example, the second voltage peak is located where the current is nearly the base line. Please explain the correlation between voltage and current, particularly when both pyro and piezo effects are claimed simultaneously.
5. Another minor issue, the scale bars in figure 1 caption are mislabeled. Some notations in Fig. 1c will be helpful for understanding what are showing in the images.

Reviewer #2 (Remarks to the Author):

In this paper, an approach converts a low thermal gradient to mechanical and electrical energy output is presented. The authors demonstrated that a strip made of two different materials with different thermal expansion coefficients can oscillate almost indefinitely under a given temperature difference (60 C and ambient temperature) and also demonstrated a self-propulsion. Their finding is quite interesting and this approach may be useful to energy conversion applications. However, the manuscript mainly describes observation only, and lacks thorough discussion on choice of materials, energy conversion efficiency, and so on. I don't believe the manuscript in the current shape is acceptable to this journal. I have some suggestions/questions in case the authors are willing to strengthen the paper.

1. The rational of material choice is not clear. Does any strip that has two different materials with different thermal expansion coefficient work the same as the present strip? With different thermal expansion, any biomorph strip will bend, and one tip will cool down when the tip is farther than the other tip.

2. What is the rationale of choosing PVDF and PDG-CNT? What's the role of ferroelectric property of PVDF?
3. Please explain what 3-D aligned PVDF means. What is the significance of alignment of PVDF. Fig S1 in SM simply shows it is beta phase, but doesn't explain any physics.
4. Define the 'alpha'. This angle is used frequently, but has never been defined in the main text.
5. Please explain how voltage and currents are measured. What's the role of PVDF and PEDOT:PSS in electricity conversion?
6. Probably calculating the mechanical or electrical energy conversion efficiency will be useful, compared to Carnot efficiency given that there are two different temperature sources. This second law efficiency doesn't need to be high, but probably will show a good sense how efficiently energy can be harvested/converted from waste heat source.
7. The predator analogy of the self-propelling structure may sound interesting, but doesn't seem appropriate in the journal. I recommend to emphasize more on rigorous and scientific analysis and discussion.

Reviewer #3 (Remarks to the Author):

In the paper "In-built thermo-mechanical cooperative feedback mechanism for self-propelled multimodal locomotion and electricity generation", X. Wang, et al. report on autonomous oscillating/rolling device based on PVDF/PDG-CNT bi-layered actuator. The device is able to oscillate and oscillate-to-roll on a hot place, without human influence, and can generate electricity from such self-motion. In my opinion, the phenomenon is very interesting, and the authors did a comprehensive study, and have nice demonstration about the device functions, such as caterpillar-mimic and electricity generation. I recommend it to be published after minor revisions below.

1. P2, line3 "without needing power input". The power input is the temperature gradient.
2. Line5 "multigait soft energy robotics"... What does this mean?
3. P3, "That is to say, unless the external stimulus is repeatedly switched on-off, they face limitations in performing repetitive macroscopic motion at high actuation speed under a constant/invariant operation mode."

There are some self oscillation papers in light responsive liquid crystal polymers and with quite high frequency, such as:

T. J. White, N. V. Tabiryan, S. V. Serak, U. A. Hrozhyk, V. P. Tondiglia, H. Koerner, R. A. Vaia, T. J. Bunning, *Soft Matter* 2008, 4, 1796. (first report)
 S. Serak, N. Tabiryan, R. Vergara, T. J. White, R. A. Vaia, T. J. Bunning, *Soft Matter* 2010, 6, 779. (271 Hz!)

These should be cited.

4. P18. "It is apparent that the TMES shows effective oscillations under non-concentrated and constant

sunlight irradiation (Supplementary Movie 12)."

The oscillation starts at 55C (from Fig.5), where the temperature gradient allows establishing mechanical feedback in the actuator. Under sunlight of 85 mWcm² intensity, how is it possible that the elevated temperature can have similar effect as 55C hot plate? Is it rather possible that outdoors, also wind/air currents play a role, especially for such small film-like devices? Also in the movie, the film is not oscillating very regularly, perhaps due to air fluctuation?

5. Similar oscillating phenomena has been reported also in, for example, in the paper "Light-Powered Tumbler Movement of Graphene Oxide/Polymer Nanocomposites" by You et al. which should be cited.

6. Such self-oscillation phenomena should not be restricted to some specific type of material, but I believe it should be quite general concept for all kinds of bi-layered actuators or stimuli-responsive materials.

Few things the authors may consider:

How about scaling up/down the device, in which case one would have to adjust between several parameters, like gravity, friction, curvature, etc. Few sentences on such prospects would be very helpful.

The electricity generation efficiency, although it will be a very small number, should be estimated, to give readers a better knowledge of this system.

Reviewers' comments:

Reviewer #1 (Remarks to the Author):

This manuscript reports an interesting development of thermo-mechano-electrical system (TMES) using a multilayered soft material structure. This device is able to perform perpetual and multimodal locomotions including translational oscillation, rotational and revolving when placed on a hot surface. This heat-induced motion is further demonstrated for electric energy generation and generating bio-mimic responses. This work is innovative and demonstrates a new way of using low-level thermal energy to achieve multiple intriguing functions. The heat-induced motion mechanism is also analyzed in great details. This work possesses a high level of novelty and quality for a publication on Nature Communications. However, there still are several issues that need to be addressed.

1. The motion of TMES was attributed to the temperature and heat transfer. The results at different surface temperature was shown and compared. Heat transfer is also related to the temperature gradient or the temperature difference. This may be largely related to how fast the ring oscillates or how fast it moves. Therefore, temperature difference between different points on the device also need to be analyzed and correlate to the motion patterns.
2. Following above question, the smaller the ring is, less temperature gradient there is. Can the authors comment and predict how this behavior changes as the size of the ring reduces? What would be the balance between the temperature and the ring size?
3. The fluorescence trace image shown in fig. 2c is confusing. Considering the ring was rolling back and forth, the lateral positions of the fluorescence spots should constantly change and this change should be more obvious for upper spots. Why in the image, all the fluorescence spot traces seem following the same profile? The device profile also need to be provided in the image to reveal the relative position changes.
4. In the energy harvesting part, the authors demonstrated a good match between the x position and current output. However, the voltage profile appears rather irregular. For example, the second voltage peak is located where the current is nearly the base line. Please explain the correlation between voltage and current, particularly when both pyro and piezo effects are claimed simultaneously.
5. Another minor issue, the scale bars in figure 1 caption are mislabeled. Some notations in Fig. 1c will be helpful for understanding what are showing in the images.

Reviewer #2 (Remarks to the Author):

In this paper, an approach converts a low thermal gradient to mechanical and electrical energy output is presented. The authors demonstrated that a strip made of two different materials with different thermal expansion coefficients can oscillate almost indefinitely under a given temperature difference (60 C and ambient temperature) and also demonstrated a self-propulsion. Their finding is quite interesting and this approach may be useful to energy conversion

applications. However, the manuscript mainly describes observation only, and lacks thorough discussion on choice of materials, energy conversion efficiency, and so on. I don't believe the manuscript in the current shape is acceptable to this journal. I have some suggestions/questions in case the authors are willing to strengthen the paper.

1. The rationale of material choice is not clear. Does any strip that has two different materials with different thermal expansion coefficients work the same as the present strip? With different thermal expansion, any biomorphic strip will bend, and one tip will cool down when the tip is farther than the other tip.

2. What is the rationale of choosing PVDF and PDG-CNT? What's the role of ferroelectric property of PVDF?

3. Please explain what 3-D aligned PVDF means. What is the significance of alignment of PVDF. Fig S1 in SM simply shows it is beta phase, but doesn't explain any physics.

4. Define the 'alpha'. This angle is used frequently, but has never been defined in the main text.

5. Please explain how voltage and currents are measured. What's the role of PVDF and PEDOT:PSS in electricity conversion?

6. Probably calculating the mechanical or electrical energy conversion efficiency will be useful, compared to Carnot efficiency given that there are two different temperature sources. This second law efficiency doesn't need to be high, but probably will show a good sense how efficiently energy can be harvested/converted from waste heat source.

7. The predator analogy of the self-propelling structure may sound interesting, but doesn't seem appropriate in the journal. I recommend to emphasize more on rigorous and scientific analysis and discussion.

Reviewer #3 (Remarks to the Author):

In the paper "In-built thermo-mechanical cooperative feedback mechanism for self-propelled multimodal locomotion and electricity generation", X. Wang, et al. report on autonomous oscillating/rolling device based on PVDF/PDG-CNT bi-layered actuator. The device is able to oscillate and oscillate-to-roll on a hot place, without human influence, and can generate electricity from such self-motion. In my opinion, the phenomenon is very interesting, and the authors did a comprehensive study, and have nice demonstration about the device functions, such as caterpillar-mimic and electricity generation. I recommend it to be published after minor revisions below.

1. P2, line3 "without needing power input". The power input is the temperature gradient.

2. Line5 "multigait soft energy robotics"... What does this mean?

3. P3, “That is to say, unless the external stimulus is repeatedly switched on-off, they face limitations in performing repetitive macroscopic motion at high actuation speed under a constant/invariant operation mode.”

There are some self oscillation papers in light responsive liquid crystal polymers and with quite high frequency, such as:

T. J. White, N. V. Tabiryan, S. V. Serak, U. A. Hrozhyk, V. P. Tondiglia, H. Koerner, R. A. Vaia, T. J. Bunning, *Soft Matter* 2008, 4, 1796. (first report)

S. Serak, N. Tabiryan, R. Vergara, T. J. White, R. A. Vaia, T. J. Bunning, *Soft Matter* 2010, 6, 779. (271 Hz!)

These should be cited.

4. P18. “It is apparent that the TMES shows effective oscillations under non-concentrated and constant sunlight irradiation (Supplementary Movie 12).”

The oscillation starts at 55C (from Fig.5), where the temperature gradient allows establishing mechanical feedback in the actuator. Under sunlight of 85 mWcm² intensity, how is it possible that the elevated temperature can have similar effect as 55C hot plate? Is it rather possible that outdoors, also wind/air currents play a role, especially for such small film-like devices? Also in the movie, the film is not oscillating very regularly, perhaps due to air fluctuation?

5. Similar oscillating phenomena has been reported also in, for example, in the paper

"Light-Powered Tumbler Movement of Graphene Oxide/Polymer Nanocomposites" by You et al. which should be cited.

6. Such self-oscillation phenomena should not be restricted to some specific type of material, but I believe it should be quite general concept for all kinds of bi-layered actuators or stimuli-responsive materials.

Few things the authors may consider:

How about scaling up/down the device, in which case one would have to adjust between several parameters, like gravity, friction, curvature, etc. Few sentences on such prospects would be very helpful.

The electricity generation efficiency, although it will be a very small number, should be estimated, to give readers a better knowledge of this system.

Reviewer #1 (Remarks to the Author):

Reviewer: *This manuscript reports an interesting development of thermo-mechano-electrical system (TMES) using a multilayered soft material structure. This device is able to perform perpetual and multimodal locomotions including translational oscillation, rotational and revolving when placed on a hot surface. This heat-induced motion is further demonstrated for electric energy generation and generating bio-mimic responses. This work is innovative and demonstrates a new way of using low-level thermal energy to achieve multiple intriguing functions. The heat-induced motion mechanism is also analyzed in great details. This work possesses a high level of novelty and quality for a publication on Nature Communications. However, there still are several issues that need to be addressed.*

Response: We are thankful to the reviewer for his/her positive assessment on the novelty and quality of our work.

Reviewer: *The motion of TMES was attributed to the temperature and heat transfer. The results at different surface temperature was shown and compared. Heat transfer is also related to the temperature gradient or the temperature difference. This may be largely related to how fast the ring oscillates or how fast it moves. Therefore, temperature difference between different points on the device also need to be analyzed and correlate to the motion patterns.*

Response: We agree with the reviewer's suggestion. The thermal fluctuation in TMES is essential to continuous locomotions. The overall thermal effect in TMES that sustains the continuous oscillation is discussed (Fig. 3b and corresponding description). In the revised paper, we have further plotted temperature profile of points on the ring in one oscillation cycle, by analyzing snapshots of the infrared video at a time resolution of 1/30 using software FLIR Tools+, which reads:

“Temperature profile of marked points on TMES is additionally analyzed (See details in Supplementary Fig. 15).” (P12, line 4-5)

Supplementary Figure 15. Temperature profile of marked TMES. (a) An infrared image of TMES with marked points on the hot surface at 60 °C. (b) Temperature profile of the marked points in one oscillation cycle.

The points near the extreme ends, 1, 2, 6 and 7, undergo thermal fluctuations in a small temperature range of 25 to 35 °C, and the highest temperature of point 3 and 4 reaches 42 °C when they contact the hot surface. The temperature of point 5 can rise to as high as 54 °C due to its relative long duration in proximity to/contact on the hot surface during the period when TMES oscillates from the medium to rightmost and then back to medium position (~ 0.6 s). (P16 in Supplementary materials)

Reviewer: Following above question, the smaller the ring is, less temperature gradient there is. Can the authors comment and predict how this behavior changes as the size of the ring reduces? What would be the balance between the temperature and the ring size?

Response: In the work, we standardized the length and width of the TMES strip at 6 cm and 1.5 cm respectively. In these dimensions, TMES strip with $\alpha = 90^\circ$ exhibits oscillations from 50 to 70 °C and self-rolling forward motion from 70 to 73 °C (Fig. 4d). TMES stay static when the temperature is lower than 50 °C, and continuous motions can't be sustained at temperatures above 73 °C. We have also demonstrated that a TMES sample in square shape (6 × 6 cm) shows similar motion characteristics to that of TMES strip (Supplementary Fig. 10). This suggests that the width (the direction along the longitudinal alignment of PVDF in TMES) does not have key impact to the motion modes. However, changing the length will most likely affect the motion modes at different temperature ranges.

Drastic oscillations of TMES strip (1.5 × 6 cm) begins around 55 °C. As discussed in the section of “Thermo-mechanical feedback mechanism for autonomous locomotions”, the geometrical shape of TMES, *i.e.*, curvature differences in the ring and temperature gradient, activate and sustain the continuous motions. With a smaller length of the strip, the TMES will be subjected to a decreased temperature gradient which leads to a higher geometrical symmetry and smaller curvature difference in the ring. As a result, a TMES strip of 4 cm in length for example, will stay static at 55 °C due to its higher geometrical symmetry and decreased heat dissipation on two ends of the ring, as illustrated in the following schematic diagram. The oscillation motion of

TMES strip 4 cm in length can start at a higher temperature. In addition, self-propelled rolling forward motion starts at approximately 70 °C when TMES strip (6 cm) oscillates to a complete circular ring with a curvature $\rho = 2\pi / L = 2\pi / (6\text{ cm}) = 1.05\text{ cm}^{-1}$. The activation of rolling motion of a TMES strip 4 cm in length will need a higher temperature because of its requirement for a larger curvature of 1.57 cm^{-1} . In summary, a smaller length of the ring will require for higher activation temperatures for oscillation and self-propelling rolling motions. In our experiment, the length of TMES is designed at 6 cm to further harvest thermo-mechanical oscillation energy for pyro/piezoelectric generation over the temperature range of 55 to 70 °C.

We have added some sentences on this prospect in the section of “self-propelled multimodal locomotions”, which reads:

“In addition, changing the length of TMES will affect the working temperature ranges of different modal locomotions. For example, if the length of TMES strip at $\alpha = 90^\circ$ decreases from 6 to 4 cm, activation temperatures of its self-oscillation and rolling motion should be higher than 50 and 70 °C, respectively. The thermo-mechanical feedback triggered motion proposed here represents a general concept for bi-layered actuators or thermo-mechanical responsive materials, and it’s possible to further design various autonomous locomotions working at different temperature ranges based on different thermo-mechanical response sensitivities of the materials systems.” (P14, line 12-19)

Reviewer: *The fluorescence trace image shown in fig. 2c is confusing. Considering the ring was rolling back and forth, the lateral positions of the fluorescence spots should constantly change and this change should be more obvious for upper spots. Why in the image, all the fluorescence spot traces seem following the same profile? The device profile also need to be provided in the image to reveal the relative position changes.*

Response: During the locomotions, the lateral profile of TMES in length direction is in an elliptical shape, and this ring oscillates from left to right. To do kinematic tracking, seven fluorescence spots were marked on the ring, and the moving trajectories of the seven points were recorded to characterize and analyze the oscillation process. As suggested by the reviewer, a photograph of marked TMES in visible light has been added as an inset in Fig. 2c to clarify the characterization process.

Figure 2| Autonomous thermo-mechanical oscillation, kinematic tracking and mechanical analysis. (a) Photograph of TMES on a hot surface at 60 °C. (b) Infrared images and curvature change of TMES on the hot surface with different temperatures. (c) Fluorescent image of marked TMES in UV light and its 2D coordinate system. The inset is a corresponding photograph of marked TMES in visible light. (d) The 2D trajectory of marked points on the cross-section of TMES during a complete oscillation cycle from the leftmost to the rightmost and then back to the leftmost position, at four different temperatures, 50, 55, 60 and 65 °C (from bottom to top). (e) Plot of center of gravity during a complete oscillation cycle based on trajectory of the marked points. (f) The 2D trajectory of center of gravity at different temperatures. (g) Time-dependent center of gravity in the x-axis during three consecutive oscillation cycles at different temperatures. All scale bars correspond to 1 cm. (P8)

Reviewer: In the energy harvesting part, the authors demonstrated a good match between the x position and current output. However, the voltage profile appears rather irregular. For example, the second voltage peak is located where the current is nearly the base line. Please explain the correlation between voltage and current, particularly when both pyro and piezo effects are claimed simultaneously.

Response: The β phase of PVDF has ferroelectric properties owing to a crystalline structure obtained by an all trans arrangement of the polymer chains that gives rise to a permanent dipole, which makes it both pyroelectric and piezoelectric nanogenerator (*Energy Environ. Sci.*, 2014, 7, 3836–3856). The pyroelectric effect relies on thermal fluctuations, which leads to the orientation change of dipoles in the material, and thus changes the spontaneous polarization. The piezoelectric effect is activated by applications of mechanical stretching-releasing/compressing-releasing, as spontaneous polarization changes when dipole moment changes in responsive to the

mechanical stimulus. Conventionally, the two effects work independently when external temperature change or mechanical stimulus acts on the material. In our previous paper, we integrated the two generating effects in a single cell using microstructure design and materials assembly (*Adv. Energy Mater.* 2015, 5, 1500704).

In this work, the thermo-mechanical feedback loop produces auto-sustained temperature fluctuations and mechanical mobility in TMES without needing external change in stimulus. As shown in Fig. 5, a good match between short circuit current (I_{sc}) and the x position indicates that the output of I_{sc} synchronizes with each motion of TMES. Meanwhile, the motion is accompanied by slight and fast mechanical bending-releasing of PVDF in TMES as shown in Supplementary Movie 10, therefore we believe that piezoelectric effect dominates the peak I_{sc} output. On the other hand, we can see that only two main open circuit voltage (V_{oc}) peaks (positive value) appear in one oscillation cycle, which is attributed to the alternative heating/cooling pyroelectric effect. The output of I_{sc} and V_{oc} are independently based on piezoelectric and pyroelectric phenomena, respectively, enabling cooperative thermal and mechanical locomotion energy harvesting. The second voltage peak (negative value) is located where the current generated is nearly at the value of the base line, when TMES stays almost static and motionless at the rightmost position but is undergoes a heating process, detected by the pyroelectric effect. However, peak I_{sc} and V_{oc} (positive values) both appear at around the time when TMES switches from the leftmost to rightmost or the rightmost to leftmost position, as the maximum rate of change of thermal and mechanical stimulus both occurring at this moment.

To clarify the special thermo-mechanical locomotion generating behavior, we have added more detailed explanations of the generation of V_{oc} in one oscillation cycle and revised the presentation of Fig. 5, which now reads and shows:

“It is observed that two V_{oc} peaks appear during one oscillation cycle, which are attributed to two positional switching and distinct thermal heating/cooling effect acting on TMES (Supplementary Movie 9). As shown in Fig. 5b, the V_{oc} continuously increases, when TMES stays almost stationary in the leftmost position while continuous air cooling effect acts on the right side of TMES. The cooling results in a continuous increase of V_{oc} until fast switching of TMES from the leftmost to rightmost position reverses the cooling to heating effect. Correspondingly, V_{oc} gradually decreases and the first V_{oc} peak emerges. After a short heating effect due to heat transfer from the hot surface to the right side of TMES, air cooling effect takes on the left side of TMES, thereupon V_{oc} keeps increasing accordingly until TMES switches from the rightmost to leftmost position, and the second V_{oc} peak appears. Therefore, the heating/cooling pyroelectric effect mainly contributes to V_{oc} output while locomotions sustain the essential cyclic thermal fluctuations. In contrast, I_{sc} peaks correspond to each of the oscillation motions (Supplementary Movie 10), suggesting that mechanical piezoelectric effect dominates the peak I_{sc} output.” (P16-17)

Figure 5| Thermomechanical locomotion energy harvesting utilizing the pyro/piezoelectric effect. (a) Time-dependent shifting in the x-axis, short circuit current and open circuit voltage of TMES ($\alpha = 90^\circ$) during five consecutive oscillation cycles on the hot surface at 60 °C and (b) magnification in one oscillation cycle. (c) Average cycling time, maximum displacement, transition speed, and Peak I_{sc} at 55, 60 and 65 °C. (P17)

Reviewer: Another minor issue, the scale bars in figure 1 caption are mislabeled. Some notations in Fig. 1c will be helpful for understanding what are showing in the images.

Response: As suggested by the reviewer, the mislabeled scale bars are corrected, and the images in Fig. 1c are captioned clearly with more details.

Figure 1| The structure design and system concept of TMES. (a) Schematic illustration of the design concept of thermo-mechano-electrical conversion based on a bimorph actuator. (b) Photograph of a large-area TMES sample. (c) Cross-sectional scanning electron microscopy (SEM) image of PVDF/PDG-CNT bimorph (left) and magnification of PDG-CNT nanocomposite layer (right). The scale in (b), left and right of (c) are 1 cm, 10 μm and 200 nm, respectively. (P4)

Reviewer #2 (Remarks to the Author):

Reviewer: *In this paper, an approach converts a low thermal gradient to mechanical and electrical energy output is presented. The authors demonstrated that a strip made of two different materials with different thermal expansion coefficients can oscillate almost indefinitely under a given temperature difference (60 C and ambient temperature) and also demonstrated a self-propulsion. Their finding is quite interesting and this approach may be useful to energy conversion applications. However, the manuscript mainly describes observation only, and lacks thorough discussion on choice of materials, energy conversion efficiency, and so on. I don't believe the manuscript in the current shape is acceptable to this journal. I have some suggestions/questions in case the authors are willing to strengthen the paper.*

Response: We thanks for the positive comments on our findings and helpful suggestions to our work. Following the reviewer's advice, we have revised our papers, clarified the selection of materials and calculated energy conversion efficiency of the thermo-mechanical locomotions.

Reviewer: *The rational of material choice is not clear. Does any strip that has two different materials with different thermal expansion coefficient work the same as the present strip? With different thermal expansion, any biomorph strip will bend, and one tip will cool down when the tip is farther than the other tip.*

Response: We believe that the thermo-mechanical feedback mechanism for self-propelled multimodal locomotion introduced here should be a general concept for bi-layered actuators or other thermo-mechanical responsive materials, and it's promising to other researchers to realize similar autonomous thermo-mechanical locomotion by similar rational design using other materials system.

Reviewer: *What is the rational of choosing PVDF and PDG-CNT? What's the role of ferroelectric property of PVDF?*

Response: In this work, the most significant achievements are the onset of self-propelled mechanical locomotions fueled solely by constant low-grade heat, and simultaneous harvesting of thermo-mechanical locomotions for continuous electric energy output. Ferroelectric poly(vinylidene difluoride) (PVDF) is a fascinating material for harvesting thermal and mechanical energy utilizing pyroelectric and piezoelectric effect, and meanwhile, mechanical flexibility and extremely high coefficient of thermal expansion (CTE, $197.3 \times 10^{-6} \text{ K}^{-1}$) makes it a good candidate for using as the active layer in a bimorph thermal actuator (*Polym. Chem.*, 2012, 3, 962–969). As a result, the choice of ferroelectric PVDF is essential for the design of bimorph actuator and its pyro/piezoelectric function further conform the prerequisite for thermal and mechanical energy harvesting. Moreover, the alignment control in PVDF is designed for formation of regular and symmetrical open ring strip and chiral shapes of thermo-mechano-electrical system (TMES) on the hot surface, which avoids otherwise possible uncontrollable thermal deformation, and provides a foundation for realization of autonomous, perpetual and multimodal mechanical locomotions. The other passive layer in the bimorph actuator should be a film with low CTE and electrically conductive to serve as one of the electrodes on ferroelectric PVDF in TMES. In this regard, graphene has a negative CTE ($-8 \times 10^{-6} \text{ K}^{-1}$) and excellent electric conductivity (*Nano Lett.* 2011, 11, 3227-3231). In this work, water-based ink is

developed to deposit the graphene film of a predefined thickness on the surface of PVDF. This is to avoid dissolution of the PVDF by organic solvents such as Dimethyl Formamide. Dopamine was used to reduce graphene oxide in the preparation process to obtain polydopamine modified reduced graphene oxide (PDG) water ink. We casted PDG ink on the PVDF and dried it at 40 °C until the desired thickness was obtained. However, we found that the PDG film was easily detached from TMES during the thermal activated locomotion, presumably due to poor adhesion between the layers in PDG. To solve this problem, we introduced a small amount of multiwalled carbon nanotube (CNT, weight ratio of 1:9 to PDG) to bind the layers and thus, PDG-CNT water-based ink is used in the fabrication of the final devices. As shown in Fig. 1c, the obtained PDG-CNT has a nacre-like brick-and-mortar microstructures, the synergistic toughening effect makes PDG-CNT mechanically robust (*ACS Nano* 2015, 9, 11568-11573), and TMES, made with the PDG-CNT film, is stable without observable delamination in the locomotion tests under the range of temperatures used in the experiments.

We agree with the reviewer suggestions that the rationale of materials selection and system design should be clear. Thus, we have revised the explanations under the section of “Autonomous thermo-mechano-electrical system design”, which now reads:

“The TMES mainly consists of a three-dimensionally (3D) aligned ferroelectric polyvinylidene fluoride (PVDF) and polydopamine modified reduced graphene oxide-carbon nanotube layer (PDG-CNT) with nacre-like brick-and-mortar microstructures³⁶. The extremely high CTE and mechanical flexibility of PVDF makes it suitable as an active layer in a bimorph thermal actuator³⁷. Moreover, the pyroelectric and piezoelectric properties of the ferroelectric β -phase are the essential prerequisite for further harvesting thermal and mechanical energy using TMES (Supplementary Fig.1)¹⁸.” (P5, line 9-15)

“Based on alignment control, TMES can form a regular and symmetrical open ring strip and chiral shapes, including various left-handed or right-handed chirality strips (Supplementary Fig. 6) on the hot surface. Notably, the 3D alignment introduces directionality to the otherwise uncontrollable thermal deformation, providing a foundation for realization of autonomous, perpetual and multimodal mechanical locomotions.” (P6, line 1-6)

“The other passive layer in the bimorph actuator should be a film with low CTE and electrically conductive to serve as one of the electrodes on ferroelectric PVDF in TMES. In this regard, graphene has a negative CTE ($-8 \times 10^{-6} \text{ K}^{-1}$)³⁸ and excellent electric conductivity. The PDG-CNT synthesized in our work possesses nacre-like brick-and-mortar microstructures (Fig. 1c), and π - π bonding and strong adhesion between PDG and CNT allows PDG-CNT water ink being layer-by-layer deposited on hydrophilic treated PVDF surface without delamination³⁹, hence ensuring mechanical robustness of TMES to bear large and various deformations.” (P6, line 8-14)

“37. Thakur, V. K. *et al.* Novel polymer nanocomposites from bioinspired green aqueous functionalization of BNNTs. *Polym. Chem.* **3**, 962-969 (2012).

38. Yoon, D., Son, Y.-W. & Cheong, H. Negative thermal expansion coefficient of graphene measured by raman spectroscopy. *Nano Letters* **11**, 3227-3231 (2011).

39. Gong, S. *et al.* Integrated ternary bioinspired nanocomposites via synergistic toughening of reduced graphene oxide and double-walled carbon nanotubes. *ACS Nano* **9**, 11568-11573

(2015).” (P25)

Reviewer: Please explain what 3-D aligned PVDF means. What is the significance of alignment of PVDF. Fig S1 in SM simply shows it is beta phase, but doesn't explain any physics.

Response: The different thermal activated deformations including open ring and chiral shapes are obtained by changing the alignment directions of PVDF in TMES (Supplementary Fig. 5 and Fig. 6), subsequently we can realize autonomous and multimodal locomotions: the open ring strip for self-oscillation and self-propelled rolling, the left-handed strip for anticlockwise rotation and the right-handed strip for clockwise rotation. The alignment does not introduce anisotropic mechanical property to PVDF, as PVDF is isotropic in terms of modulus (Supplementary Fig. 3). Using an optical microscope, we observed linear textures on the surface of PVDF. Moreover, PVDF along the transverse alignment direction exhibited obviously larger expansion than that in the longitudinal direction when the hot surface temperature increased from 23 to 60 °C (Supplementary Fig. 4). Such a phenomenon was also reported in the previous work (*J. Am. Chem. Soc.* 2016, 138, 225–230), the authors embedded aligned CNT in paraffin wax, and the paraffin wax/CNT showed anisotropic thermal expansion. We thought the surface alignment of PVDF is not enough to produce the anisotropic thermal expansion. Then we carefully prepared PVDF samples for SEM measurements. SEM images in Supplementary Fig. 2 in the revised manuscript confirm that lamellar orientation exists both on the surface and cross-section. That's why we name the PVDF as 3D aligned PVDF. The β phase of PVDF has ferroelectric properties owing to a crystalline structure obtained by an all trans arrangement of the polymer chains that gives rise to a permanent dipole, which makes it both a pyroelectric and piezoelectric nanogenerator (*Energy Environ. Sci.*, 2014, 7, 3836–3856).

In the revised manuscript, we have added SEM images and discussions to clarify the 3D alignment and β phase as follows:

“The 3D alignment was confirmed by the existence of lamellar orientation both on the surface and cross-section of the PVDF (Supplementary Fig. 2), and the PVDF is isotropic in terms of modulus of ~1656 MPa (Supplementary Fig. 3) but anisotropic in regards to CTE, showing large transverse thermal expansion and small longitudinal thermal expansion (See details in Supplementary Fig. 4). A direction characterized by anticlockwise angular offset α is defined here as the angle between the longitudinal alignment direction of PVDF and the length direction of TMES strip (Supplementary Fig. 5). Based on alignment control, TMES can form a regular and symmetrical open ring strip and chiral shapes, including various left-handed or right-handed chirality strips (Supplementary Fig. 6) on the hot surface. Notably, the 3D alignment introduces directionality to the otherwise uncontrollable thermal deformation, providing a foundation for realization of autonomous, perpetual and multimodal mechanical locomotions.” (P5-6)

Supplementary Figure 2. Surface (a) and cross-sectional (b) scanning electron microscopy (SEM) image of 3D aligned PVDF film. The scale bar in (a) and (b) are 50 and 10 μm , respectively. (P3 in Supplementary Materials)

“The β phase of PVDF has ferroelectric properties owing to a crystalline structure obtained by an all trans arrangement of the polymer chains. This gives rise to a permanent dipole¹, which makes it both a pyroelectric and piezoelectric nanogenerator.” (P2 in Supplementary Materials)

- “1. Bowen, C. R. *et al.* Pyroelectric materials and devices for energy harvesting applications. *Energy Environ. Sci.* **7**, 3836-3856 (2014).” (P27 in Supplementary Materials)

Reviewer: Define the ‘alpha’. This angle is used frequently, but has never been defined in the main text.

Response: As suggested by the reviewer, we have defined α in the main text in the section of “Autonomous thermo-mechano-electrical system design”.

“A direction characterized by anticlockwise angular offset α is defined here as the angle between the longitudinal alignment direction of PVDF and the length direction of TMES strip (Supplementary Fig. 5).” (P5-6)

Reviewer: Please explain how voltage and currents are measured. What’s the role of PVDF and PEDOT:PSS in electricity conversion?

Response: The β phase of PVDF has ferroelectric properties owing to a crystalline structure obtained by an all trans arrangement of the polymer chains. This gives rise to a permanent dipole, which makes it both a pyroelectric and piezoelectric nanogenerator (*Energy Environ. Sci.*, 2014, **7**, 3836–3856). The pyroelectric effect relies on thermal fluctuations, which leads to the orientation change of dipoles in the material, and thus changes the spontaneous polarization. The piezoelectric effect is activated by applications of mechanical stretching-releasing/compressing-releasing, as spontaneous polarization changes when dipole moment changes in responsive to mechanical stimulus. The pyroelectric and piezoelectric functions of ferroelectric PVDF are utilized here to harvest thermal and mechanical locomotion energy. The PDG-CNT layer serves

as the top electrode on PVDF ($\alpha = 90^\circ$), the other side of PVDF is coated with thin PEDOT:PSS as the bottom electrode, and carbon fibers are finally attached to the two faces at the edge (Supplementary Fig. 19). The currents were collected by connecting the two carbon fibers to a low-noise current preamplifier (MODEL SR570) in a short-circuit mode, and the voltages were collected by connecting to an electrometer (KEITHLEY 6517B) in an open-circuit mode. The results were recorded by an oscilloscope (Tektronix DPO 7254). The corresponding descriptions in the revised manuscript now read:

“The PDG-CNT layer serves as the top electrode on PVDF ($\alpha = 90^\circ$), the other side is coated with thin poly(3,4-ethylenedioxythiophene)-poly(styrenesulfonate) (PEDOT:PSS, $\sim 1 \mu\text{m}$) as the bottom electrode, and carbon fibers are finally attached to the two faces at the edge.” (P16, line 5-8)

“The output of short circuit currents and open circuit voltages during the thermo-mechanical locomotion process were measured by connecting the two electrodes of TMES to a low-noise current preamplifier (MODEL SR570) and an electrometer (KEITHLEY 6517B), respectively, and the results were recorded by an oscilloscope (Tektronix DPO 7254).” (P22, line 18-22)

***Reviewer:** Probably calculating the mechanical or electrical energy conversion efficiency will be useful, compared to Carnot efficiency given that there are two different temperature sources. This second law efficiency doesn't need to be high, but probably will show a good sense how efficiently energy can be harvested/converted from waste heat source.*

Response: As suggested by the reviewer, we have added energy conversion efficiency calculation in the revised manuscript as follows:

“Calculated energy conversion efficiencies of TMES

With regards to the energy conversion efficiency, we postulate that the energy conversion to be as such:

Waste heat energy \rightarrow Gain in gravitational potential energy (due to deformation and raising of centroid) \rightarrow Kinetic energy and electrical energy

Energy conversion efficiency η is,

$$\frac{\text{Gain in gravitational potential energy}}{\text{Total heat input}}$$

To estimate the total heat input, we assume the hot surface is an ideal blackbody of 60°C , having surface area of 1.5 cm by 6 cm. Using Stefan-Boltzmann law, the power provided by the hot surface is 0.6287 W or 0.6916 J in one oscillatory cycle of 1.1s.

The potential energy of the system is mgh , approximately $2.5 \times 10^{-5}\text{J}$, where the centroid of TMES is raised by 12.5 mm and the weight is 0.2 g.

The efficiency of energy conversion is approximately 0.004 %.

The conversion of energy from gravitational potential energy to kinetic energy and electrical energy is more efficient. The kinetic energy consists of both the translational and rotational kinetic energy. The translational kinetic energy ($\frac{1}{2}mv^2$), computed from the lateral speed of the centroid of TMES (88.7mm/s) is 7.868×10^{-7} J. The rotational kinetic energy ($\frac{1}{2}I\omega^2$) is computed as shown below.

Assuming that TMES is an incomplete circular cylinder,

Radius of curvature: 11.1 mm

$$I = mR^2$$

$$\rightarrow I \approx 24.642 \text{ g}\cdot\text{mm}^2$$

$$\omega = \frac{v}{r} \approx 8 \text{ rad/s}$$

$$\text{Rotational kinetic energy} \approx 1.577 \times 10^{-6} \text{ J}$$

Taking the integral of the voltage and current ($\int_{t_0}^{t_1} V \cdot Idt$), where t_0 and t_1 are start and end time of one oscillatory cycle respectively, the total electrical energy generated is 6.94×10^{-7} J.

The efficiency of conversion from potential energy to kinetic and electrical energy is approximately 12.2 %.” (P26 in Supplementary Materials)

In addition, the Carnot efficiency for the system is

$$\frac{T_H - T_C}{T_H} \times 100\% = \frac{333.15 - 298.15}{333.15} \times 100\% \approx 10.5\%$$

Reviewer: *The predator analogy of the self-propelling structure may sound interesting, but doesn't seem appropriate in the journal. I recommend to emphasize more on rigorous and scientific analysis and discussion.*

Response: In this work, we propose a thermo-mechano-electrical system (TMES) which is based on the thermo-mechanical feedback mechanism and pyro/piezoelectric function design. In the section of “Thermo-mechanical feedback mechanism for autonomous locomotions”, we did a comprehensive study including kinematic tracking, mechanical analysis, and dynamic thermal imaging to disclose and explain the thermo-mechanical feedback principle. In the revised manuscript, we have analyzed the thermo-mechanical locomotion generating effect in more detail, and explained how the open circuit voltages and short circuit currents generate based on pyroelectric and piezoelectric theory (P16-17). Finally, a prototypical soft robot termed TMES-bot was designed to showcase the functionality of our device, such as bioinspired self-defensive locomotion and electricity generation. We expect the demonstration can inspire more thinking

and idea on the further exploration of applications of our thermo-mechano-electrical system in multi-disciplinary research fields.

Reviewer #3 (Remarks to the Author):

Reviewer: *In the paper “In-built thermo-mechanical cooperative feedback mechanism for self-propelled multimodal locomotion and electricity generation”, X. Wang, et al. report on autonomous oscillating/rolling device based on PVDF/PDG-CNT bi-layered actuator. The device is able to oscillate and oscillate-to-roll on a hot place, without human influence, and can generate electricity from such self-motion. In my opinion, the phenomenon is very interesting, and the authors did a comprehensive study, and have nice demonstration about the device functions, such as caterpillar-mimic and electricity generation. I recommend it to be published after minor revisions below.*

Response: We thank the reviewer for his/her positive assessment.

Reviewer: *P2, line3 “without needing power input”. The power input is the temperature gradient.*

Response: In this work, we aim at exploring untapped low grade heat below 100 °C for multiple functions, such as multimodal locomotions and energy generations. The temperature gradient that attenuates from the hot surface to above air activates and maintains the locomotions of our thermo-mechano-electrical system (TMES). Liberated from conventional electrical, pneumatic or light controls, TMES can work on a hot place in a wide temperature range over 50 to 80 °C, and probably these hot surfaces can be some heating equipment surfaces, or sunlight naturally heated surfaces in ambient environment which has been demonstrated in our work (Fig. 6e and Supplementary Movie 12). In this context, our TMES can autonomously utilize this kind of waste heat and further extract it for electric power generation without needing electrical power input.

Reviewer: *Line5 “multigait soft energy robotics”... What does this mean?*

Response: Multigait means multiple patterns of movement or locomotions. The TMES is capable of executing multimodal locomotions and simultaneously generate electric energy. Hence we have used “multigait soft energy robotics” to describe our work.

Reviewer: *P3, “That is to say, unless the external stimulus is repeatedly switched on-off, they face limitations in performing repetitive macroscopic motion at high actuation speed under a constant/invariant operation mode.”*

There are some self oscillation papers in light responsive liquid crystal polymers and with quite high frequency, such as:

T. J. White, N. V. Tabiryan, S. V. Serak, U. A. Hrozhyk, V. P. Tondiglia, H. Koerner, R. A. Vaia, T. J. Bunning, Soft Matter 2008, 4, 1796. (first report)

S. Serak, N. Tabiryan, R. Vergara, T. J. White, R. A. Vaia, T. J. Bunning, Soft Matter 2010, 6, 779. (271 Hz!)

These should be cited.

Response: As suggested by the reviewer, we have cited these two papers in the revised manuscript, which now reads:

“Also, except for several light activated self-oscillating materials³³⁻³⁵, most of these systems rely on or tether to external circuits and/or control devices to regulate or on-off the external stimulus for continuous cyclical motion.” (P3, line 8-10)

“33. White, T. J. *et al.* A high frequency photodriven polymer oscillator. *Soft Matter* **4**, 1796-1798 (2008).

34. Serak, S. *et al.* Liquid crystalline polymer cantilever oscillators fueled by light. *Soft Matter* **6**, 779-783 (2010)” (P25)

Reviewer: P18. “It is apparent that the TMES shows effective oscillations under non-concentrated and constant sunlight irradiation (Supplementary Movie 12).”

The oscillation starts at 55C (from Fig.5), where the temperature gradient allows establishing mechanical feedback in the actuator. Under sunlight of 85 mWcm2 intensity, how is it possible that the elevated temperature can have similar effect as 55C hot plate? Is it rather possible that outdoors, also wind/air currents play a role, especially for such small film-like devices? Also in the movie, the film is not oscillating very regularly, perhaps due to air fluctuation?

Response: In the outdoor experiment on a sunny and unwindy day, TMES strip was placed on the surface of a black tile pavement which was heated up naturally by the sunlight, and time-dependent solar intensity was monitored while the corresponding ambient temperature and surface temperature of the black tiled were recorded. Supplementary movie 12 and energy generation data were collected when the temperature of the black tiled and air were around 52 and 31 °C, respectively (Supplementary Fig. 23). Different from the indoor condition with hot surface temperature at 55 °C and air temperature at 23 °C, temperature gradient was smaller in outdoor condition because of the hotter atmosphere. Higher temperature of the air contributes to larger bending deformation of TMES (higher geometrical instability), and thus TMES can lose its mechanical balance easier and start dramatic oscillation motions on a hot surface at a lower temperature (52 °C). However, the smaller temperature gradient will lower its oscillation speed and stability because of its attenuated thermal heating/cooling effects. In addition, the outdoor hot surface temperature also slightly fluctuated due to the fluctuation of solar intensity. Hence, the oscillations under outdoor condition seem not that regular.

Reviewer: *Similar oscillating phenomena has been reported also in, for example, in the paper "Light-Powered Tumbler Movement of Graphene Oxide/Polymer Nanocomposites" by You et al. which should be cited.*

Response: As suggested by the reviewer, we have cited this paper in the revised manuscript, which now reads:

“Also, except for several light activated self-oscillating materials³³⁻³⁵, most of these systems rely on or tether to external circuits and/or control devices to regulate or on-off the external stimulus for continuous cyclical motion.” (P3, line 8-10)

“35. Yu, L. & Yu, H. Light-powered tumbler movement of graphene oxide/polymer nanocomposites. *ACS Appl. Mater. Interfaces* **7**, 3834-3839 (2015).” (P25)

Reviewer: *Such self-oscillation phenomena should not be restricted to some specific type of material, but I believe it should be quite general concept for all kinds of bi-layered actuators or stimuli-responsive materials.*

Response: We agree with the reviewer’s opinion. The thermo-mechanical feedback mechanism for the autonomous motions proposed here is a general concept for bi-layered actuators or some other thermo-mechanical responsive materials.

Reviewer: *Few things the authors may consider:*

How about scaling up/down the device, in which case one would have to adjust between several parameters, like gravity, friction, curvature, etc. Few sentences on such prospects would be very helpful.

Response: In the work, we standardized the length and width of the TMES strip at 6 cm and 1.5 cm respectively. In these dimensions, TMES strip with $\alpha = 90^\circ$ exhibits oscillations from 50 to 70 °C and self-rolling forward motion from 70 to 73 °C (Fig. 4d). TMES stay static when the temperature is lower than 50 °C, and continuous motions can’t be sustained at temperatures above 73 °C. We have also demonstrated that a TMES sample in square shape (6 × 6 cm) shows similar motion characteristics to that of TMES strip (Supplementary Fig. 10). This suggests that the width (the direction along the longitudinal alignment of PVDF in TMES) does not have key impact to the motion modes. However, changing the length will most likely affect the motion modes at different temperature ranges.

Drastic oscillations of TMES strip (1.5 × 6 cm) begin at around 55 °C. As discussed in the section of “Thermo-mechanical feedback mechanism for autonomous locomotions”, the geometrical shape of TMES, *i.e.*, curvature differences in the ring and temperature gradient, activate and sustain the continuous motions. With a smaller length of the strip, the TMES will be subjected to a decreased temperature gradient which leads to a higher geometrical symmetry and smaller curvature difference in the ring. As a result, a TMES strip of 4 cm in length for example, will stay static at 55 °C due to its higher geometrical symmetry and decreased heat dissipation on two ends of the ring, as illustrated in the following schematic diagram. The oscillation motion of TMES strip 4 cm in length can start at a higher temperature. In addition, self-propelled rolling forward motion starts at approximately 70 °C when TMES strip (6 cm) oscillates to a complete circular ring with a curvature $\rho = 2\pi / L = 2\pi / (6\text{ cm}) = 1.05\text{ cm}^{-1}$. The activation of rolling motion of a TMES strip 4 cm in length will need a higher temperature because of its requirement for a larger curvature of 1.57 cm^{-1} . In summary, a smaller length of the ring will require for higher activation temperatures for oscillation and self-propelling rolling motions. In our experiment, the length of TMES is designed at 6 cm to further harvest thermo-mechanical oscillation energy for pyro/piezoelectric generation over the temperature range of 55 to 70 °C.

As suggested by the reviewer, we have added some discussion on the prospect in the section of “self-propelled multimodal locomotions”, which reads:

“In addition, changing the length of TMES strip will affect the working temperature ranges of different modal locomotions. For example, if the length of TMES strip at $\alpha = 90^\circ$ decreases from 6 to 4 cm, activation temperatures of its self-oscillation and rolling motion should be higher than 50 and 70 °C, respectively. The thermo-mechanical feedback triggered motion proposed here represents a general concept for bi-layered actuators or other thermo-mechanical responsive materials, and it’s possible to further design various autonomous locomotions working at different temperature ranges based on different thermo-mechanical response sensitivities of the materials systems.” (P14, line 12-19)

Reviewer: *The electricity generation efficiency, although it will be a very small number, should be estimated, to give readers a better knowledge of this system.*

Response: To give the readers a better knowledge of this system, we have discussed the energy conversion process in the revised manuscript as follows:

“Calculated energy conversion efficiencies of TMES

With regards to the energy conversion efficiency, we postulate that the energy conversion to be as such:

Waste heat energy → Gain in gravitational potential energy (due to deformation and raising of centroid) → Kinetic energy and electrical energy

Energy conversion efficiency η is,

$$\eta = \frac{\text{Gain in gravitational potential energy}}{\text{Total heat input}}$$

To estimate the total heat input, we assume the hot surface is an ideal blackbody of 60 °C, having surface area of 1.5 cm by 6 cm. Using Stefan-Boltzmann law, the power provided by the hot surface is 0.6287 W or 0.6916 J in one oscillatory cycle of 1.1s.

The potential energy of the system is mgh , approximately $2.5 \times 10^{-5} \text{J}$, where the centroid of TMES is raised by 12.5 mm and the weight is 0.2 g.

The efficiency of energy conversion is approximately 0.004 %.

The conversion of energy from gravitational potential energy to kinetic energy and electrical energy is more efficient. The kinetic energy consists of both the translational and rotational kinetic energy. The translational kinetic energy $\left(\frac{1}{2}mv^2\right)$, computed from the lateral speed of the centroid of TMES (88.7mm/s) is $7.868 \times 10^{-7} \text{J}$. The rotational kinetic energy $\left(\frac{1}{2}I\omega^2\right)$ is computed as shown below.

Assuming that TMES is an incomplete circular cylinder,

Radius of curvature: 11.1 mm

$$I = mR^2$$

$$\rightarrow I \approx 24.642 \text{ g}\cdot\text{mm}^2$$

$$\omega = \frac{v}{r} \approx 8 \text{ rad/s}$$

$$\text{Rotational kinetic energy} \approx 1.577 \times 10^{-6} \text{ J}$$

Taking the integral of the voltage and current $\left(\int_{t_0}^{t_1} V \cdot Idt\right)$, where t_0 and t_1 are start and end time of one oscillatory cycle respectively, the total electrical energy generated is $6.94 \times 10^{-7} \text{J}$.

The efficiency of conversion from potential energy to kinetic and electrical energy is approximately 12.2 %.” (P26 in *Supplementary Materials*)

REVIEWERS' COMMENTS:

Reviewer #1 (Remarks to the Author):

In the revised manuscript, the authors completely addressed all my concerns with great details. This is a very nice piece of work with a good technological significance and scientific depth. I would recommend publication of this manuscript in its current form.

Reviewer #2 (Remarks to the Author):

The authors have addressed most of my concerns well. My only suggestion is below:

1. In the page 13 of the rebuttal, "The pyroelectric effect relies on thermal fluctuations, which leads to the orientation change of dipoles in the material, and thus changes the spontaneous polarization. The piezoelectric effect is activated by applications of mechanical stretching-releasing/compressing-releasing, as spontaneous polarization changes when dipole moment changes in responsive to mechanical stimulus. The pyroelectric and piezoelectric functions of ferroelectric PVDF are utilized here to harvest thermal and mechanical locomotion energy."

This explanation is helpful. In the manuscript, the authors simply state that mechanical and electrical energy can be obtained due to pyroelectric/piezoelectric properties of PVDF, which doesn't deliver the physics to the readers clearly.

Probably more intuitive and straightforward explanation will be much more beneficial to readers. Therefore, please consider to include a paragraph or several phrases, for example, temperature difference (wasteheat and air) leads to bending  creates mechanical stress and high temperature gradient due to cooling in the air  mechanical stress generate electricity (piezo) and temperature gradient (and fluctuation due to rolling) generates electricity (pyro).

Reviewer #3 (Remarks to the Author):

The manuscript was already strong in the originally submitted form, and seeing the rigorous addressing of the reviewers' comments, I am confident to recommend acceptance of the paper in its present form. It is well conducted, well presented, and demonstrates a novel approach to thermomechanical materials that is likely to trigger interest in the scientific community.

I would, however, like to point out, that in terms of motility under constant stimulus and autonomous actuation, both of which are important attributes for this paper, previous works on liquid-crystal polymers have been recently published in this same journal [NComm 2016-7-12260 - Photomotility of Polymers; NComm 2017-8-15546 - A Light-Driven Artificial Flytrap]. I recommend the authors to have a look at these papers and judge whether they consider them relevant enough to be added to the citation list. Finally, ref 7 & 32 is a duplicate.

Congratulations for a job well done!

REVIEWERS' COMMENTS:

Reviewer #1 (Remarks to the Author):

In the revised manuscript, the authors completely addressed all my concerns with great details. This is a very nice piece of work with a good technological significance and scientific depth. I would recommend publication of this manuscript in its current form.

Reviewer #2 (Remarks to the Author):

The authors have addressed most of my concerns well. My only suggestion is below:

1. In the page 13 of the rebuttal, “The pyroelectric effect relies on thermal fluctuations, which leads to the orientation change of dipoles in the material, and thus changes the spontaneous polarization. The piezoelectric effect is activated by applications of mechanical stretching-releasing/compressing-releasing, as spontaneous polarization changes when dipole moment changes in responsive to mechanical stimulus. The pyroelectric and piezoelectric functions of ferroelectric PVDF are utilized here to harvest thermal and mechanical locomotion energy.”

This explanation is helpful. In the manuscript, the authors simply state that mechanical and electrical energy can be obtained due to pyroelectric/piezoelectric properties of PVDF, which doesn't deliver the physics to the readers clearly.

Probably more intuitive and straightforward explanation will be much more beneficial to readers. Therefore, please consider to include a paragraph or several phrases, for example, temperature difference (wasteheat and air) leads to bending  creates mechanical stress and high temperature gradient due to cooling in the air  mechanical stress generate electricity (piezo) and temperature gradient (and fluctuation due to rolling) generates electricity (pyro).

Reviewer #3 (Remarks to the Author):

The manuscript was already strong in the originally submitted form, and seeing the rigorous addressing of the reviewers' comments, I am confident to recommend acceptance of the paper in its present form. It is well conducted, well presented, and demonstrates a novel approach to thermomechanical materials that is likely to trigger interest in the scientific community.

I would, however, like to point out, that in terms of motility under constant stimulus and autonomous actuation, both of which are important attributes for this paper, previous works on liquid-crystal polymers have been recently published in this same journal [NComm 2016-7-12260 - Photomotility of Polymers; NComm 2017-8-15546 - A Light-Driven Artificial Flytrap]. I recommend the authors to have a look at these papers and judge whether they consider them relevant enough to be added to the citation list. Finally, ref 7 & 32 is a duplicate.

Congratulations for a job well done!

Reviewer #1 (Remarks to the Author):

In the revised manuscript, the authors completely addressed all my concerns with great details. This is a very nice piece of work with a good technological significance and scientific depth. I would recommend publication of this manuscript in its current form.

Response: We greatly appreciate the reviewer for his/her positive assessment and recommendation of publishing our work in *Nature Communications*.

Reviewer #2 (Remarks to the Author):

The authors have addressed most of my concerns well. My only suggestion is below:

1. In the page 13 of the rebuttal, “The pyroelectric effect relies on thermal fluctuations, which leads to the orientation change of dipoles in the material, and thus changes the spontaneous polarization. The piezoelectric effect is activated by applications of mechanical stretching-releasing/compressing-releasing, as spontaneous polarization changes when dipole moment changes in responsive to mechanical stimulus. The pyroelectric and piezoelectric functions of ferroelectric PVDF are utilized here to harvest thermal and mechanical locomotion energy.”

This explanation is helpful. In the manuscript, the authors simply state that mechanical and electrical energy can be obtained due to pyroelectric/piezoelectric properties of PVDF, which doesn't deliver the physics to the readers clearly.

Probably more intuitive and straightforward explanation will be much more beneficial to readers. Therefore, please consider to include a paragraph or several phrases, for example, temperature difference (wasteheat and air) leads to bending  creates mechanical stress and high temperature gradient due to cooling in the air  mechanical stress generate electricity (piezo) and temperature gradient (and fluctuation due to rolling) generates electricity (pyro).

Response: We greatly appreciate the reviewer for his/her helpful suggestions to improve our work. As suggested by the reviewer, we have added some phrases to explain the principle of thermo-mechano-electrical system (TMES) in detail as follows:

“In the thermo-mechano-electrical system developed here, the temperature difference between waste heat and air triggers mechanical deformation of TMES, and also sustains the continuous locomotions and temperature fluctuations. The mechanical locomotion stress and temperature gradient are correspondingly harvested as piezoelectricity and pyroelectricity based on the ferroelectric property of PVDF.” (P18)

Reviewer #3 (Remarks to the Author):

The manuscript was already strong in the originally submitted form, and seeing the rigorous addressing of the reviewers' comments, I am confident to recommend acceptance of the paper in its present form. It is well conducted, well presented, and demonstrates a novel approach to thermomechanical materials that is likely to trigger interest in the scientific community.

Response: We greatly appreciate the reviewer for his/her positive assessment and kind recommendation of publishing our work in *Nature Communications*.

I would, however, like to point out, that in terms of motility under constant stimulus and autonomous actuation, both of which are important attributes for this paper, previous works on liquid-crystal polymers have been recently published in this same journal [NComm 2016-7-12260 - Photomotility of Polymers; NComm 2017-8-15546 - A Light-Driven Artificial Flytrap]. I recommend the authors to have a look at these papers and judge whether they consider them relevant enough to be added to the citation list. Finally, ref 7 & 32 is a duplicate.

Response: The reviewer is right. As suggested, the two excellent work related to autonomous actuation of liquid-crystal polymers have been cited as ref 31 (*Nat. Commun.* **2016**, 7, 13260) and ref 32 (*Nat. Commun.* **2017**, 8, 15546) in the revised manuscript, respectively. We have deleted the duplicate of ref 7.

Congratulations for a job well done!

Response: We greatly thank for the reviewer.

Once again, many thanks to the editors and referees for the profound comments and suggestions on improving the quality of our work, and we are very grateful for your time and effort in the processing of our manuscript.

Yours sincerely, Ho Ghim Wei